# $\beta$-DPO: Direct Preference Optimization with Dynamic $\beta$

**Junkang Wu**[1][*] **Yuexiang Xie**[2] **Zhengyi Yang**[1] **Jiancan Wu**[1][†]
**Jinyang Gao**[2] **Bolin Ding**[2] **Xiang Wang**[1] **Xiangnan He**[1][†]
[1]University of Science and Technology of China
[2]Alibaba Group
{jkwu0909, yangzhy1998, wujcan, xiangwang1223, xiangnanhe}@gmail.com,
{yuexiang.xyx, jinyang.gjy, bolin.ding}@alibaba-inc.com

## Abstract

Direct Preference Optimization (DPO) has emerged as a compelling approach for training Large Language Models (LLMs) to adhere to human preferences. However, the performance of DPO is sensitive to the fine-tuning of its trade-off parameter $\beta$, as well as to the quality of the preference data. We analyze the impact of $\beta$ and data quality on DPO, uncovering that optimal $\beta$ values vary with the informativeness of pairwise data. Addressing the limitations of static $\beta$ values, we introduce a novel framework that dynamically calibrates $\beta$ at the batch level, informed by data quality considerations. Additionally, our method incorporates $\beta$-guided data filtering to safeguard against the influence of outliers. Through empirical evaluation, we demonstrate that our dynamic $\beta$ adjustment technique significantly improves DPO's performance across a range of models and datasets, offering a more robust and adaptable training paradigm for aligning LLMs with human feedback. The code is available at https://github.com/junkangwu/beta-DPO.

## 1 Introduction

The alignment of Large Language Models (LLMs) with human feedback, as explored in works like GPT-4 and LLaMA-2 [29, 38, 8], has marked a significant advancement in generating responses that are more helpful, factual, and ethical [30]. Among the various alignment strategies, Reinforcement Learning from Human Feedback (RLHF) [30] is a notable method that refines LLMs using the Proximal Policy Optimization (PPO) algorithm [36]. This approach employs a KL divergence penalty to ensure minimal deviation of the model from its original configuration, ensuring the retention of its initial characteristics while improving alignment.

Despite the effectiveness, RLHF's instability and computational requirements often limit its practical applications, prompting the exploration of alternatives like Direct Preference Optimization (DPO) [34]. DPO circumvents the reinforcement learning loop by exploiting the inherent connection between reward functions and optimal policies, thereby simplifying the policy model training. It encourages the model to favor the response that aligns with human preferences ($\mathbf{y}_w$) over the dispreferred ($\mathbf{y}_l$), implying DPO's sensitivity to the quality of pairwise data. The balance between maintaining the original reference model ($\pi_{\text{ref}}$) and incorporating new preferences ($\pi_{\boldsymbol{\theta}}$) is controlled by a $\beta$ hyperparameter, whose lower values advocate for aggressive updates and higher values support more

---

[*]Work done at Alibaba Group.
[†]Jiancan Wu and Xiangnan He are the corresponding authors.

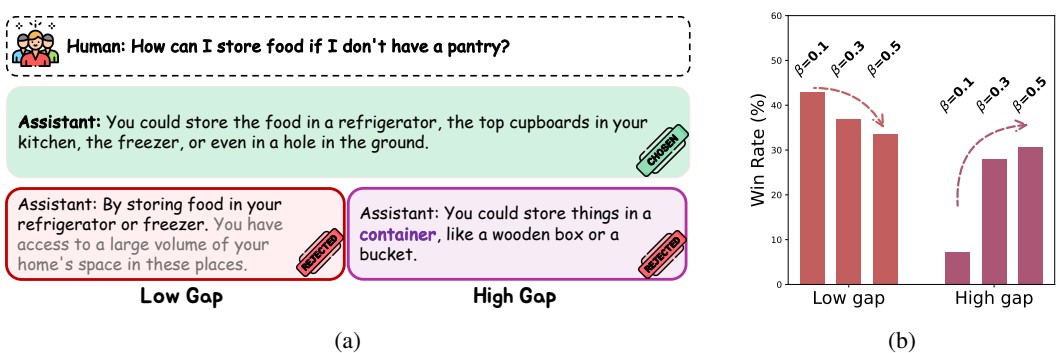

Figure 1: (**1a**) **Pairwise Data: Low vs. High Gap**: "Low gap" denotes cases where the chosen and rejected examples are closely similar, typically indicating high-quality, informative pairs. "High gap" signifies pairs with larger differences, implying lower-quality data. (**1b**) **Influence of Data Quality on $\beta$ Selection:** Pythia-1.4B's performance on the HH dataset reveals a distinct trend: for "Low gap", a higher $\beta$ reduces win rate, whereas for "High gap", an increased $\beta$ improves it.

conservative adjustments:

$$\ell_{\text{DPO}}(\boldsymbol{\theta}) = \mathbb{E}_{\mathbf{x}, \mathbf{y}_w, \mathbf{y}_l} [-\log \sigma (\beta [\log (\frac{\pi_{\boldsymbol{\theta}}(\mathbf{y}_w|\mathbf{x})}{\pi_{\text{ref}}(\mathbf{y}_w|\mathbf{x})}) - \log (\frac{\pi_{\boldsymbol{\theta}}(\mathbf{y}_l|\mathbf{x})}{\pi_{\text{ref}}(\mathbf{y}_l|\mathbf{x})})])].$$

However, the current DPO literature has largely overlooked the joint influence of $\beta$ selection and pairwise data $(\mathbf{y}_w, \mathbf{y}_l)$'s quality on the alignment performance. To bridge this gap, we conduct a preliminary experiment to investigate how different $\beta$ selections influence the model performance under two distinct data pair gap conditions, as shown in Figure 1b. In *low gap* scenarios (*cf.* Figure 1a), where the difference between preferred and dispreferred samples is minor, an increase in $\beta$ (*e.g.,* from 0.1 to 0.5) corresponds with a decline in win rate (*e.g.,* from 42% to 33%). Conversely, in *high gap* situations (*cf.* Figure 1a) where a significant difference exists, an increase in $\beta$ tends to improve DPO performance. Such contrasting outcomes highlight the necessity to tailor the $\beta$ value contingent upon the data quality, especially in the presence of outliers [4].

Building upon these insights and recognizing the mixture nature of diverse quality in practically collected data [31], we propose a dynamic $\beta$ selection strategy for DPO. This strategy adaptively adjusts $\beta$ in response to the quality of pairwise data and robustifies DPO against data variability. Intuitively, one straightforward solution is personalizing $\beta$ for each pair, rather than fixing it across the population of all pairs. While conceptually appealing, (1) instance-level $\beta$ personalization can lead to optimization instabilities, particularly when dealing with a vast array of human preference instances (*cf.* Section 5.3). This issue underscores the challenge of balancing $\beta$ updates with stable and scalable DPO. In this light, we propose to explore a *batch-level* adaptation of $\beta$, aiming to balance update aggressiveness and training stability. Moreover, (2) the frequent occurrence of outliers necessitates a strategy for accurately adjusting the batch-level $\beta$. The dataset notably features outliers, a challenge underscored by the significant reward discrepancy variations within the training samples of the dataset (*cf.* Section 4.1). Such conditions impede the model's ability to accurately estimate the batch-level $\beta$, thereby undermining the effectiveness of batch-level $\beta$ calibration. To this end, we propose a simple yet effective dual-component approach:

- **Dynamic $\beta$ Calibration at Batch-Level** (for Challenge 1): To mitigate optimization instabilities, we dynamically calibrate $\beta$ within each batch. Specifically, this batch-level adjustment is based on data quality, with $\beta$ being adaptively decreased for high-quality, closely-matched pairwise data (*i.e., low gap* data) to facilitate assertive updates. While for easily-discriminated pairs (*i.e., high-gap* data), $\beta$ is increased to promote cautious updates, preventing overfitting to noise. This targeted batch-level calibration enables stable and responsive optimization.

- **$\beta$-Guided Data Filtering** (for Challenge 2): We implement a $\beta$-guided data filtering approach to tackle the frequent occurrence of outliers. By establishing a benchmark $\beta$ value for filtering incoming data at the batch level, we maintain the fidelity of $\beta$ estimation by prioritizing the most reliable and representative samples. As a result, it diminishes the impact of outliers that might

otherwise derail the optimization process, thus enhancing the precision and robustness of the batch-level $\beta$ calibration.

Our contributions are threefold: (1) We investigate a pioneering study on the joint influence of $\beta$ selection and pairwise data quality on the DPO performance. (2) We introduce a simple yet effective $\beta$-dynamic strategy for DPO, adaptively balancing the update aggressiveness and training stability. (3) Through empirical evaluations, our approach demonstrates marked improvements in performance across diverse conditions and model sizes (*e.g.,* achieving improvements exceeding 10% on models of various sizes, including Pythia-410M, 1.4B, and 2.8B).

## 2    Related Work

**Reinforcement Learning from Human Feedback.** Despite RLHF's effectiveness in aligning language models (LMs) with human values [10, 3, 38, 30, 14, 15], its complexity and resource demands have spurred the exploration of alternatives. RAFT [12] selects optimal training samples via an existing reward model, whereas RRHF [45] employs a simpler ranking loss, retaining PPO's efficiency. Diverse from these, DPO [34] directly optimizes LMs using a preference-based loss function, showcasing enhanced training stability in comparison to traditional RLHF. Innovatively, SLiC-HF [47] and KTO [13] devise loss functions rooted in human decision-making, focusing on preference calibration and utility optimization, respectively. Dr. DPO [42] consider robust settings where safety or group information is known at training time. Further, RSO [25] and ORPO [19] introduce efficient preference modeling and optimization, with ORPO uniquely combining supervised fine-tuning and preference alignment. These advancements reflect the ongoing shift towards more efficient, nuanced RL methods.

**Data Quality in LLM's Alignment.** Recent studies have increasingly recognized the significance of data quality in the alignment of LLMs. For example, LIMA [48] leverages heuristics such as post scores, response lengths, formatting, and topics to manually craft 1000 high-quality datasets from StackExchange, wikiHow, and Reddit for superficial alignment. In a similar vein, Bai et al. [3] prioritize data points based on user engagement levels for dataset assembly. Rejection Sampling (RS) and Best-of-$N$ (BoN) techniques, as evidenced in the works of Nakano et al. [28] and Gao et al. [16], involve selecting the optimal candidate from $N$ generated possibilities through the application of a reward model. To enhance preference optimization, RSO [25] uses statistical weightings to differentiate outcomes from an optimal policy and a base SFT policy. Besides, fDPO [27] employs a Reward Model to filter out low-quality data, effectively addressing dataset quality concerns.

## 3    Preliminaries

Given a text sequence (commonly referred to as a prompt) $\mathbf{x}$, a sequence $\mathbf{y} = [y_1, y_2, \ldots y_N]$ is generated as a response to the prompt $\mathbf{x}$. An autoregressive language model $\pi$, when provided with the prompt $\mathbf{x}$, can generate the response sequence $\mathbf{y}$ following the probability decomposition:

$$\pi(\mathbf{y}|\mathbf{x}) = \prod_{t=1}^{N} \pi(y_i|\mathbf{x}, \mathbf{y}_{<t}), \tag{1}$$

where $\mathbf{y}_{<t}$ denotes the preceding tokens in the response sequence. Now, given a preference dataset $\mathcal{D} = \{(\mathbf{x}^{(i)}, \mathbf{y}_w^{(i)}, \mathbf{y}_l^{(i)})\}_{i=1}^{M}$, wherein each triplet consists of a prompt $\mathbf{x}$ with two responses $\mathbf{y}_w \in \Sigma^*$ and $\mathbf{y}_l \in \Sigma^*$, with $\Sigma^*$ representing the alphabet, a preference oracle — either a human annotator or a language model — provides preference feedback $o(\mathbf{y}_w \succ \mathbf{y}_l|\mathbf{x}) \in \{0, 1\}$, indicating whether $\mathbf{y}_w$ is preferred over $\mathbf{y}_l$. We denote $\mathbb{P}(\mathbf{y}_w \succ \mathbf{y}_l|\mathbf{x}) = \mathbb{E}[o(\mathbf{y}_w \succ \mathbf{y}_l|\mathbf{x})]$ the probability of $\mathbf{y}_w$ "winning the duel" over $\mathbf{y}_l$. The Kullback-Leibler (KL) divergence between two probability distributions with densities $p$ and $q$ is defined as $\text{KL}(p\|q) = \mathbb{E}_{\mathbf{y} \sim p(\mathbf{y})}\left[\log \frac{p(\mathbf{y})}{q(\mathbf{y})}\right]$.

**RLHF with Reward Models.** Christiano et al. [10] pioneer the learning of a reward function $r(\mathbf{y}; \mathbf{x})$ based on the Bradley-Terry model [7]. This model is deployed for the triplet of a prompt $(\mathbf{x})$ and two responses $(\mathbf{y}_w, \mathbf{y}_l)$, establishing the likelihood of preference for $\mathbf{y}_w$ over $\mathbf{y}_l$ as:

$$\mathbb{P}(\mathbf{y}_w \succ \mathbf{y}_l|\mathbf{x}) = \frac{\exp(r(\mathbf{y}_w; \mathbf{x}))}{\exp(r(\mathbf{y}_w; \mathbf{x})) + \exp(r(\mathbf{y}_l; \mathbf{x}))} = \sigma\big(r(\mathbf{y}_w; \mathbf{x}) - r(\mathbf{y}_l; \mathbf{x})\big), \tag{2}$$

where $\sigma(x) = e^x/(e^x + 1)$ represents the logistic function. The approach for estimating the reward function within the Bradley-Terry framework is to maximize the log-likelihood $\log \mathbb{P}(\mathbf{y}_w \succ \mathbf{y}_l | \mathbf{x})$. Assuming accurate estimation of the true reward function $r(\mathbf{y}; \mathbf{x})$, Christiano et al. [10] propose to solve the following problem with policy optimization algorithms in RL such as PPO [36]:

$$\max_{\boldsymbol{\theta}} \mathbb{E}_{\mathbf{x} \sim \mathcal{X}, \mathbf{y} \sim \pi_{\boldsymbol{\theta}}(\cdot | \mathbf{x})}[r(\mathbf{y}; \mathbf{x})] - \beta \mathbb{E}_{\mathbf{x} \sim \mathcal{X}}[\mathrm{KL}(\pi_{\boldsymbol{\theta}}(\cdot | \mathbf{x}) \| \pi_{\mathrm{ref}}(\cdot | \mathbf{x}))], \tag{3}$$

where $\mathcal{X}$ represents the prompt distribution, $r(\mathbf{y}; \mathbf{x})$ denotes the reward function learned using the Bradley-Terry model on the preference dataset, $\pi_{\mathrm{ref}}$ is the fixed reference model (typically selected to be the one post supervised fine-tuning), and $\beta$ serves as the penalty coefficient of the KL divergence.

**Directed Preference Optimization (DPO).** Rafailov et al. [34] identify that the optimization problem above has a closed-form solution such that for any $\mathbf{y}$,

$$\pi^*(\mathbf{y}|\mathbf{x}) \propto \pi_{\mathrm{ref}}(\mathbf{y}|\mathbf{x}) \exp(r(\mathbf{y}; \mathbf{x})/\beta),$$

which can be further converted to the DPO loss for any triplet $(\mathbf{x}, \mathbf{y}_w, \mathbf{y}_l)$:

$$\ell_{\mathrm{DPO}}(\mathbf{x}, \mathbf{y}_w, \mathbf{y}_l; \boldsymbol{\theta}; \pi_{\mathrm{ref}}) = -\log \sigma \left( \beta \left[ \log \left( \frac{\pi_{\boldsymbol{\theta}}(\mathbf{y}_w|\mathbf{x})}{\pi_{\mathrm{ref}}(\mathbf{y}_w|\mathbf{x})} \right) - \log \left( \frac{\pi_{\boldsymbol{\theta}}(\mathbf{y}_l|\mathbf{x})}{\pi_{\mathrm{ref}}(\mathbf{y}_l|\mathbf{x})} \right) \right] \right). \tag{4}$$

## 4 Method

In this section, we investigate the critical connection between the parameter $\beta$ and the quality of pairwise data in optimizing DPO. We present empirical evidence demonstrating the effect of $\beta$ settings on DPO performance across datasets of varying quality. Our proposed method, $\beta$-DPO, introduces dynamic calibration of $\beta$ and a data filtering mechanism tailored to improve DPO's effectiveness across diverse data conditions.

### 4.1 Motivation: The Impact of Pairwise Data Quality on $\beta$ Selection

Scrutinizing Equation (4), we argue that DPO's effectiveness critically hinges on two factors: the choice of $\beta$ and the quality of pairwise data. Here, we conduct experiments to demonstrate the influence of variations in $\beta$ and data quality on DPO, pivotal for its effective real-world application.

**Datasets.** We utilize the Anthropic HH dataset [3] for our experimental analysis, which contains approximately 170,000 dialogues between humans and an automated assistant. In this dataset, a human inquiry, denoted as $\mathbf{x}$, is paired with two responses $(\mathbf{y}_w, \mathbf{y}_l)$, where $\mathbf{y}_w$ represents the response favored by the human annotator, while $y_l$ is the alternate response. Notably, the alternate response $y_l$ retains informational value, making this dataset high-quality with minimal discrepancies between the response pairs, which we classify as a *low gap* dataset. To further explore the impact of data quality on DPO, we construct a synthetic dataset, referred to as the *high gap* dataset. This dataset differs from the *low gap* dataset by introducing a greater disparity between responses. Specifically, the alternative response $y_l$ is generated by a Supervised FineTuned (SFT) Pythia-2.8B model, while the preferred response $y_w$ remains consistent with the original dataset. We also combine the two datasets in equal proportion to create a *mixed gap* dataset, with each contributing 50%, to incorporate the characteristics of both the *low gap* and *high gap* datasets.

**Models and Metrics.** Our study evaluates various model sizes, specifically Pythia-410M, Pythia-1.4B, and Pythia-2.8B [5], to ensure a comprehensive assessment. Following the established protocol in DPO [34], each model iteration undergoes a single epoch with a batch size of 64. This setup provides a uniform basis for evaluation across different models. We adopt the evaluation strategy from DPO [34] to calculate the *win rate*, a metric that measures how often the GPT-4 model prefers a response generated by our models over the default chosen response on the subset of the test dataset.

**Findings**: **(1) The optimal value of $\beta$ varies with data quality, reflecting divergent performance patterns across datasets.** In Figure 2, we present the win rate results across three levels of pairwise data gap, each evaluated under varying $\beta$ parameters. As can be observed, with *low gap* pairwise data, a smaller $\beta$ value is preferable for optimizing performance. This is likely

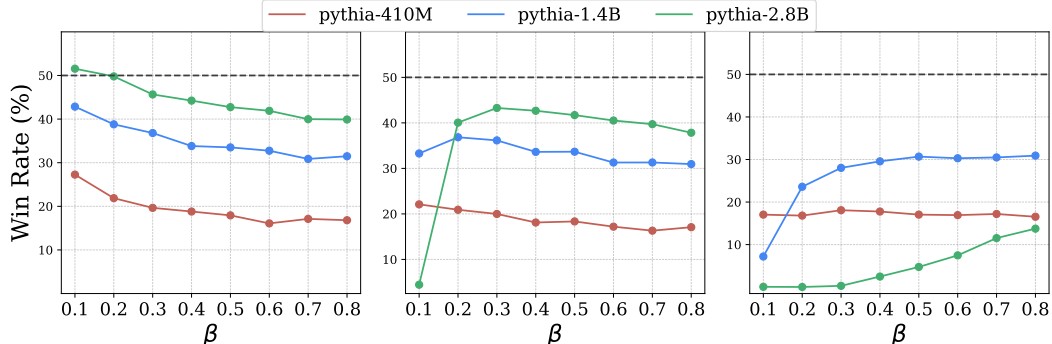

Figure 2: Win rate performance of DPO across different $\beta$ settings on the *low gap*, *mixed gap*, and *high gap* datasets.

because the informative content of such data allows a lower $\beta$ to facilitate more substantial updates, thereby enhancing alignment accuracy. Conversely, for *high gap* pairwise data, maintaining a low $\beta$ may lead to overfitting, which significantly undermines the alignment process. The *mixed gap* dataset — a combination of both *low gap* and *high gap* datasets — exhibits a more nuanced performance pattern, suggesting the necessity for a dynamic $\beta$ calibration strategy to adapt to varying data quality. Consequently, adhering to a fixed $\beta$ value, *i.e.,* configuring $\beta$ at the population level, might be inadequate for the dynamic and varied nature of real-world datasets.

**(2) The dataset exhibits notable outliers.** In Figure 3, utilizing the Pythia-2.8B model, we evaluate the data quality by examining the distribution of reward discrepancy for each triplet (which we will define as "individual reward discrepancy" later) within the HH dataset's training samples. The tails of the density plot extend beyond the highlighted percentiles, suggesting the existence of data samples with significantly higher or lower reward discrepancies. Notably, cases with significantly higher rewards for positive samples over negative ones suggest low informational value, as these discrepancies likely do not contribute meaningfully to the model's learning process. Whereas the opposite cases hint at potential labeling errors. Both cases deviate from an expected rational distribution range and are thus classified as outliers. For further details on outliers, kindly refer to Appendix A.2.

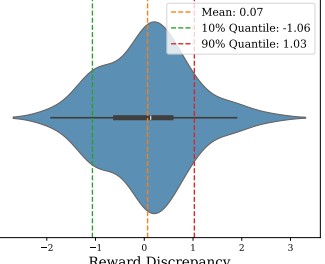

Figure 3: The distribution of individual reward discrepancy $(r(\mathbf{y}_w^{(i)}; \mathbf{x}^{(i)}) - r(\mathbf{y}_l^{(i)}; \mathbf{x}^{(i)}))$ on the training dataset of HH.

## 4.2 Method: Dynamic $\beta$ Calibration in DPO

Through our empirical analysis, we highlight the sensitivity of DPO to $\beta$ selections and the frequent occurrence of outliers. Hence, determining the optimal $\beta$ value requires careful consideration of the quality of pairwise data while also addressing the influence of outliers. This prompts the question: *what criteria define a superior choice of $\beta$?* In response, we propose the following guiding principles:

**Principle 1:** *The optimal $\beta$ value should be responsive to pairwise data's quality.*

**Principle 2:** *The selection of $\beta$ value should minimize the influence of outliers.*

### 4.2.1 Dynamic $\beta$ Calibration at Batch-Level

We begin by introducing the concept termed '*individual reward discrepancy*', which represents the difference between the rewards of winning and losing for each triplet, serving as a measurement for pairwise data quality. Formally, for a triplet $(\mathbf{x}^{(i)}, \mathbf{y}_w^{(i)}, \mathbf{y}_l^{(i)}) \in \mathcal{D}$, the individual reward discrepancy is defined as $M_i = r(\mathbf{y}_w^{(i)}; \mathbf{x}^{(i)}) - r(\mathbf{y}_l^{(i)}; \mathbf{x}^{(i)})$. While our primary analysis utilizes the implicit reward model induced by the policy trained using DPO, we also conducted comparative experiments with an explicit reward model. The details of these experiments with the explicit RM can be found in Appendix A.5. For the DPO-based implicit reward model, the reward discrepancy is expressed as:

$$M = \beta_0 \log \left( \frac{\pi_\theta(y_w \mid x)}{\pi_{\text{ref}}(y_w \mid x)} \right) - \beta_0 \log \left( \frac{\pi_\theta(y_l \mid x)}{\pi_{\text{ref}}(y_l \mid x)} \right).$$

Here, $\pi_\theta$ represents the policy being optimized, and $\pi_{\text{ref}}$ denotes the reference policy. This formulation captures the difference in the log-probabilities of the winning and losing outcomes, weighted by the parameter $\beta$. Motivated by our guiding principles, a straightforward approach is to assign a distinct $\beta$ to each triplet, allowing each $\beta$ to serve as a parameter tailored to its respective triplet. This instance-level dynamic $\beta$ adaption can be formulated as follows:

$$\beta_i = \beta_0 + \alpha(M_i - M_0)\beta_0 = [1 + \alpha(M_i - M_0)]\beta_0, \tag{5}$$

where $\beta_0$ represents the benchmark hyperparameter intrinsic to DPO, typically set to 0.1. The term $M_0$ denotes a predetermined threshold, and the coefficient $\alpha$ is a scaling factor within the interval $[0, 1]$ that adjusts the influence of $M_i$ on $\beta_i$. Specifically, when $\alpha = 0$, $\beta_i$ remains constant at $\beta_0$, thus maintaining the standard DPO framework without modification.

Equation (5) illustrates that $\beta_i$ increases monotonically with $M_i$, allowing the model to adjust the $\beta$ value based on the running reward differential between paired samples. Nevertheless, such instance-level adjustments may introduce instabilities during training. Prior studies have shown that a minibatch approach can help avoid saddle points or local minima [17], as well as mitigate the impact of noise [35, 6]. Drawing inspiration from these benefits, we propose a batch-level dynamic estimation methodology for $\beta$:

$$\beta_{\text{batch}} = [1 + \alpha(\mathbb{E}_{i \sim \text{batch}}[M_i] - M_0)]\beta_0. \tag{6}$$

In practical applications, the threshold $M_0$ can be estimated by employing the global mean of $M_i$ with a moving average updating scheme [24]:

$$M_0 \leftarrow mM_0 + (1 - m)\mathbb{E}_{i \sim \text{batch}}[M_i], \tag{7}$$

where $m \in [0, 1)$ is a momentum coefficient. Such a batch-level calibration method introduces only one new parameter, $\alpha$, to control the scale of $\beta$ adjustment. The calculation of $\mathbb{E}_{i \sim \text{batch}}[M_i]$ is straightforward within DPO procedures, thereby incurring no additional computational overhead.

### 4.2.2 $\beta$-Guided Data Filtering

To mitigate the adverse impact of outliers on the $\beta$ selection process, we introduce a $\beta$-guided data filtering mechanism. Informed by $3\sigma$ confidence criterion [33], this strategy employs a probabilistic model to assess the significance of each triplet $(\mathbf{x}^{(i)}, \mathbf{y}_w^{(i)}, \mathbf{y}_l^{(i)})$ based on its individual reward discrepancy $M_i$, which is defined as:

$$p(M_i) = \frac{1}{\sqrt{2\pi}\sigma} \exp \left( -\frac{(M_i - M_0)^2}{2\sigma^2} \right), \tag{8}$$

where $M_0$ and $\sigma$ represent the mean and standard deviation of $M_i$ across the training dataset, respectively. Similar to the updating scheme of $M_0$ in Equation (7), we dynamically estimate the value of $\sigma$ using the moving average method:

$$\sigma \leftarrow m\sigma + (1 - m)\sqrt{\mathbb{V}_{i \sim \text{batch}}[M_i]}. \tag{9}$$

This probabilistic weighting discerns the relative importance of each sample, guiding the selection of $|\text{batch}| \times \rho$ samples (without replacement) based on their calculated probabilities $p(M_i)$. Here, $\rho$ denotes the selection ratio, defaulting to 0.8, a choice validated by preliminary experiments aimed at optimizing training efficiency and model accuracy.

This process is iterated for each training batch, ensuring that the training data is continuously updated to reflect the most informative samples. The introduction of the $\beta$-guided data filtering strategy is instrumental in fortifying the model against outliers, thereby facilitating the accurate estimation of the $\beta$ value.

**Highlights:** We underline the following key features of our proposed $\beta$-DPO framework:

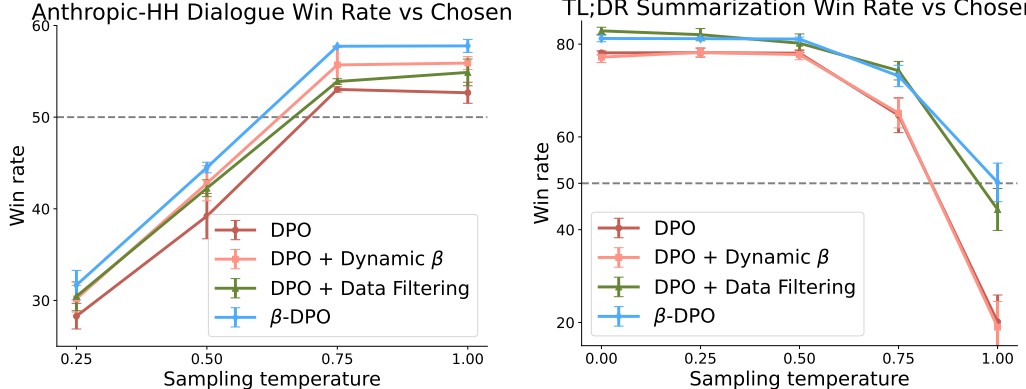

Figure 4: **Left.** The win rates computed by GPT-4 evaluations for the Anthropic-HH one-step dialogue; $\beta$-DPO consistently outperforms across all sampling temperatures. **Right.** In the comparison of TL;DR summarization win rates versus chosen summaries with GPT-4 as the evaluator, $\beta$-DPO is distinguished as the only strategy achieving a win rate over 50% across different sampling temperatures.

- **Simplicity**: $\beta$-DPO is extremely straightforward and quick to implement. It merely involves a dynamic $\beta$ adjustment at the batch level and a $\beta$-guided data filtering mechanism, both of which are predicated upon the reward discrepancy denoted by $M_i$.

- **Efficiency**: Unlike other methodologies [25, 27, 32, 40] that necessitate an additional gold model for data filtering, our method leverages the running reward discrepancy $M_i$ within the DPO framework. Moreover, our empirical studies indicate that $\beta$-DPO exhibits insensitivity to the hyperparameters $\rho$. A default setting of $\rho = 0.8$ typically yields satisfactory performance.

- **Model-agnostic**: As a variant of the traditional DPO, the proposed $\beta$-DPO can function as a plug-and-play module. It allows straightforward integration of future enhancements and extensions within the DPO framework. Our empirical investigations corroborate this assertion.

### 4.3 Discussion with Previous Studies

**Relations to Data Selection.** An increasing volume of works [25, 27, 48, 32, 44] have underscored the impact of data quality on the performance of LLM's alignment. A common practice among these efforts involves employing a so-called "gold model" for data selection. This approach, however, introduces significant computational demands and the choice of the gold model directly influences the resultant system's performance. The focus of this work, it should be noted, is not to propose a superior strategy for data selection. Instead, we aim to enhance adaptability to the quality of data by dynamically adjusting the $\beta$ parameter. This adjustment facilitates improved $\beta$ estimation by selecting data based on the reward. Moreover, Section 5.2 illustrates the compatibility of dynamic $\beta$ adjustment with other data selection methodologies.

**Relations to Recent Temperature Schemes.** Dynamic temperature frameworks have been introduced in the realm of contrastive learning, motivated by various objectives, such as addressing out-of-distribution tasks [46] or accommodating long-tail data distributions [23]. The work most closely related to ours, MACL [20], has indeed proposed an alignment-adaptive strategy; however, its primary aim was to navigate the uniformity-tolerance dilemma. Hence, the integration of dynamic temperature mechanisms with LLM's alignment remains an underexplored area against this backdrop.

## 5 Experiments

In this section, we commence by conducting an empirical evaluation of $\beta$-DPO on two specific tasks: dialogue generation and summarization. Subsequently, we analyze the various adaptations of the proposed method $\beta$-DPO. Concluding this section, we underscore the imperative need for batch-level dynamic $\beta$ calibration, highlighting its significance in the context of our study.

Table 1: Win rate comparison of Pythia-410M, -1.4B, and -2.8B models on the Anthropic HH dataset, evaluated using GPT-4.

| Method | 410M | 1.4B | 2.8B |
|---|---|---|---|
| DPO | 26.19 | 42.78 | 51.51 |
| DPO + Dynamic $\beta$ | $27.15^{+3.67\%}$ | $43.51^{+1.71\%}$ | $55.19^{+7.14\%}$ |
| DPO + Data Filtering | $29.03^{+10.84\%}$ | $46.99^{+9.84\%}$ | $53.42^{+3.71\%}$ |
| $\beta$-DPO | $30.18^{+15.23\%}$ | $48.67^{+13.77\%}$ | $57.07^{+10.79\%}$ |

## 5.1 Empirical Evaluation of $\beta$-DPO on Dialogue Generation and Summarization

**Datasets and Setup.** Our experiments are conducted on the Anthropic HH dataset [3] and Reddit TL;DR summarization dataset [39]. The training configuration follows from Rafailov et al. [34]. The goals of these experiments are to study: 1) How $\beta$-DPO performs on single-turn dialogue generation and summarization tasks; 2) How the sampling temperature affects the performance of $\beta$-DPO; 3) How $\beta$-DPO works with different model sizes. For detailed experimental settings, please refer to Appendix A.1.

**Baselines.** In our comparison, we examine the performance of $\beta$-DPO relative to its counterparts: the standard DPO, DPO implemented with a dynamic $\beta$ yet devoid of $\beta$-guided data filtering, and DPO complemented by data filtering with $\beta$ fixed at 0.1.

**Win Rate Across different Sampling Temperature.** An analysis of win rates derived from GPT-4 evaluations on the Anthropic-HH one-step dialogue demonstrates that $\beta$-DPO consistently outperforms across all sampling temperatures, as depicted in Figure 4 (Left). Furthermore, for the TL;DR summarization task, $\beta$-DPO stands out as the only approach achieving win rates above 50% for diverse sampling temperatures, which is visually represented in Figure 4 (Right). The data also suggests that while both dynamic $\beta$ and data filtering enhance DPO's effectiveness, the impact of data filtering is especially pronounced in the summarization task, likely due to the inherently greater noise present in the Reddit TL;DR summarization dataset. Notably, $\beta$-DPO exhibits a remarkable degree of robustness to variations in sampling temperature. As the temperature incrementally escalates from 0.0 to 1.0, the win rate for standard DPO plunges to a mere 25%, whereas $\beta$-DPO maintains a commendable performance level with a win rate of 54%.

**Win Rate Across Different Model Sizes.** We further evaluate the performance of $\beta$-DPO on the Anthropic HH dataset with Pythia-410M, -1.4B, and -2.8B models. The results are summarized in Table 1. We observe that $\beta$-DPO consistently outperforms DPO, DPO with dynamic $\beta$, and DPO with data filtering across all model sizes. We observe that in a smaller model, the improvement of data filtering is more significant, while in a larger model, the improvement of dynamic $\beta$ is more significant. We attribute this to the fact that the larger model has more capacity to learn the optimal policy, while the smaller model needs more help from the data filtering.

## 5.2 Adaptations of $\beta$-DPO

In this section, our inquiry is twofold: first, we aim to understand the performance of $\beta$-DPO when applied across various filtering strategies; second, we examine its efficacy across different adaptations of the DPO framework. In terms of filtering strategies, prevailing studies [32, 44] in the domain largely employ a gradient-based approach. We propose to extend this methodology into three distinct scenarios. This involves arranging the gradients of pairwise data within a batch and consequently: (1) Excluding the top 20% of samples, hereby referred to as **Filter Head**, (2) Excluding the bottom 20% of samples, hereby referred to as **Filter Tail**, (3) Excluding both the top and bottom 10% of samples, a method we denote as **Filter Tail & Head**. For a fair comparison, we maintain the amount of data excluded at 20% for the above strategies. Second, we integrate three variants of DPO into our analysis: the IPO [2], a novel approach that facilitates learning directly from preferences without the need for the Bradley-Terry (BT) model. Additionally, we consider the KTO [13], which focuses on discerning whether a preference is desirable or undesirable and SPPO [43], which approximates the Nash equilibrium. For detailed settings, we refer the reader to the supplementary material.

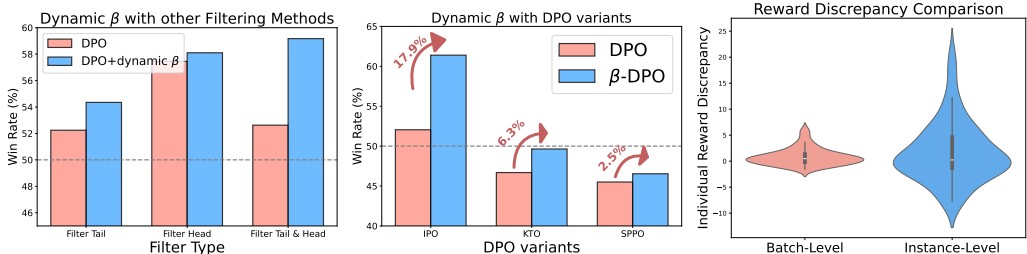

Figure 5: **Left:** Win rates from GPT-4 evaluations on Anthropic-HH single-turn dialogues, showcasing $\beta$-DPO's adaptability to diverse filtering strategies. **Middle:** Win rates of $\beta$-DPO across various DPO variants as evaluated by GPT-4. **Right:** Distribution of individual reward discrepancies following fine-tuning through batch-level and instance-level calibration.

**Selective filtering of the top 20% of samples markedly enhances model performance.** This approach, detailed in Figure 5 (Left), not only surpasses other filtering strategies but also suggests that these samples, which exhibit the smallest discrepancies between positive and negative pairs, are particularly prone to flipped noise. By excluding them, the model's learning efficacy is appreciably improved.

**Dynamic $\beta$ adapts to and improves upon existing filtering strategies.** Figure 5 (Left) corroborates our stance that a static $\beta$ proves insufficient within the DPO framework. We contend that the application of our dynamic $\beta$-DPO could markedly reshape the DPO field by fostering the development of advanced filtering techniques.

**Dynamic $\beta$ Enhancement across DPO Variants.** We introduce dynamic $\beta$-DPO, a novel strategy enhancing DPO and its variants: IPO, KTO, and SPPO in Figure 5 (Middle). Our results on the Anthropic HH dataset demonstrate that while IPO initially leads in performance, the integration of dynamic $\beta$ substantially elevates all variants, notably increasing $\beta$-IPO's efficiency by 17.9%. This underscores dynamic $\beta$-DPO's capability to significantly enhance model training through adaptable improvements, solidifying its value in advancing language models via human feedback.

### 5.3 Necessity of Batch-Level Dynamic $\beta$ Calibration

In this section, we aim to underscore the pivotal role of batch-level tuning in calibrating the parameter $\beta$. To this end, we compare the performance of our $\beta$-DPO algorithm under two distinct regimes: one employing batch-level dynamic $\beta$ calibration, and the other utilizing instance-level dynamics. To emulate the diverse data disparity scenarios encountered in practical applications, we adopt the methodology outlined in Section 4.1, meticulously blending datasets characterized by both *low gap* and *high gap* attributes at varying ratios.

**Batch-level calibration surpasses both instance-level and population-level approaches.** The results presented in Table 2 illustrate that batch-level dynamic $\beta$ calibration yields superior performance compared to instance-level dynamics and the baseline population-level approach (referred to as vanilla DPO) across a range of mixture ratios. This improvement can be credited to the batch-level calibration's ability to adjust to the varying data quality present within a batch, thus refining the model's learning process. Conversely, instance-level dynamics can provoke excessively vigorous model updates, precipitating a decline in performance particularly noticeable at a mixture ratio of 40%, a scenario in which outliers exert a significant negative impact.

**Instance-level calibration magnifies the impact of outliers.** As demonstrated in Figure 5 (Right), instance-level calibration can inadvertently widen the range of reward discrepancy distribution. This broadened range suggests that instance-level calibration might be leading to excessively high or low $\beta$ values for the model. Such disparities in $\beta$ values consequently cause disproportionate update rates for certain samples, further intensifying the extremities in the distribution.

**Our $\beta$-calibration strategy consistently outperforms baseline methods.** To expand our approach to more diverse datasets and model sizes, we follow the current state-of-the-art models SimPO [26]. We perform $\beta$-DPO with two families of models, Llama3-8B [1], Mistral2-7B [21], on the UltraChat-200k

Table 2: Comparison of win rates across varying mixture ratios on the Anthropic HH dataset, with each ratio indicating the proportion of *high-gap* to *low-gap* datasets, e.g., a 40% mixture ratio reflects a blend of 40% *high-gap* and 60% *low-gap*.

| Mixture Ratio | 10% | 20% | 30% | 40% |
|---|---|---|---|---|
| Vanilla DPO | 50.17 | 50.56 | 47.95 | 29.15 |
| + Instance-level calibration | $49.18^{-1.97\%}$ | $49.82^{-1.46\%}$ | $44.42^{-7.36\%}$ | $16.82^{-42.30\%}$ |
| + Batch-level calibration | $57.68^{+14.69\%}$ | $56.15^{+11.06\%}$ | $51.25^{+6.88\%}$ | $34.92^{+19.79\%}$ |

dataset [11]. For comparison with baselines, we assess our models using one of the most popular open-ended instruction-following benchmarks: AlpacaEval 2. All settings are consistent with SimPO [26].

Table 3 presents the AlpacaEval 2 results under the Mistral-Instruct (7B) and Llama3-Instruct (8B) settings. LC and WR denote length-controlled and raw win rate, respectively. Regardless of whether we use Llama3-8B or Mistral-7B, and whether the loss function is DPO or SimPO, our $\beta$-D, SimPO strategy consistently demonstrates significant performance improvements. This thoroughly showcases the method's strong generalization ability and excellent scalability.

Table 3: Performance comparison of different models

| Model | Mistral-Instruct (7B) | | Llama3-Instruct (8B) | |
|---|---|---|---|---|
| | LC (%) | WR (%) | LC (%) | WR (%) |
| DPO | 20.98 | **21.60** | 40.44 | 37.38 |
| $\beta$-DPO | **23.56** | 20.42 | **43.38** | **38.21** |
| SimPO | 28.50 | 30.56 | 44.38 | 38.97 |
| $\beta$-SimPO | **30.48** | **32.13** | **46.03** | **40.18** |

## 6 Conclusion and Future Work

This paper introduces $\beta$-DPO, a novel framework designed to optimize DPO by dynamically adjusting the $\beta$ parameter in response to the variability in the informativeness of pairwise data. Our approach, which incorporates $\beta$-guided data filtering and batch-level dynamic $\beta$ calibration, has demonstrated significant improvements in DPO's performance across a range of models and datasets. The empirical evaluations indicate that $\beta$-DPO offers an adaptable training paradigm for LLMs with human feedback.

**Limitations and Future Work.** Our work on $\beta$-DPO showcases a promising framework for LLM optimization, albeit with room for advancement. Future endeavors should explore: Adaptive $\beta$ in Self-Play: Extending $\beta$-DPO to self-play scenarios [9, 43] where negative samples dynamically adapt, necessitating iterative $\beta$ adjustments, to foster the evolution of superior model strategies. Enhanced Evaluation Standards: Development of sophisticated metrics and use of advanced evaluators beyond win rates, capitalizing on advancements like GPT-4+, to comprehensively gauge model performance. Scalability Investigation: Examining $\beta$-DPO's scalability to ultra-large models surpassing 7B parameters, and its integration into diverse DPO-inspired architectures, is pivotal for practical impact. Automated Parameter Tuning: Pursuing automation in parameter tuning, alleviating manual intervention for $\beta$, to streamline the training pipeline and broaden accessibility.

## Acknowledgments and Disclosure of Funding

This research is supported by the National Science and Technology Major Project (2023ZD0121102), National Natural Science Foundation of China (92270114, 62302321). This research was also supported by the advanced computing resources provided by the Supercomputing Center of the USTC.

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

# A Experiment

## A.1 $\beta$-DPO Implementation Details and Hyperparameters

$\beta$-DPO is relatively straightfoward to implement; The full algorithm is summarized in Algorithm 1.

---

**Algorithm 1** $\beta$-Direct Preference Optimization

---

**Require:** Preference dataset $\mathcal{D}$, batch size $b$, constraint coefficient $\beta_0$, selection ratio $\rho$, scaling factor $\alpha$, and learning rate $\eta$.
1: Initialize model $\pi_{\theta_0}$ with supervised finetuning on $\mathcal{D}$.
2: **while** not converged **do**
3:     Sample a batch $\mathcal{B} = \{(\mathbf{x}^{(i)}, \mathbf{y}_w^{(i)}, \mathbf{y}_l^{(i)})\}_{i=1}^b$ from $\mathcal{D}$.
4:     Compute the individual reward discrepancy $M_i = r(\mathbf{y}_w^{(i)}; \mathbf{x}^{(i)}) - r(\mathbf{y}_l^{(i)}; \mathbf{x}^{(i)})$.
5:     Update the threshold $M_0$ and $\sigma$ using Equations (7) and (9).
6:     Sample $b \times \rho$ samples without replacement based on $p(M_i)$ in Equation (8).
7:     Compute the batch-level $\beta$ using Equation (5).
8:     Compute the loss using the Equation (4).
9:     Compute the gradient and update the model $\theta_t \leftarrow \theta_{t-1} - \eta \nabla_\theta \ell(\theta_{t-1}, \mathcal{B})$.
10: **end while**
11: **return** Final model $\pi_\theta$.

---

Unless noted otherwise, we use a $\beta = 0.1$, batch size of 64, $m = 0.9$ to ensure the stability of the global $M_i$ estimation, $\rho = 0.8$ to filter 20% uninformative samples, and the Adam optimizer with a learning rate $5e - 7$ by default. We carried out all computational tasks on a suite of four 80GB A100 GPUs. For the Pythia-410M model, we use a batch size of 128, while the rest of the parameters remain the same.

In examining the arena of **single-turn dialogue**, our experimental framework leverages Pythia-2.8, Pythia-1.4b, and Pythia-410M for empirical analysis using the Anthropic-HH dataset. Given the absence of a pre-existing Supervised Fine-Tuning (SFT) model tailored for this dataset, we fine-tune an accessible language model exclusively with preferred completions to sculpt our SFT model. Turning our focus to the domain of **summarization**, we employed the Reddit TL;DR summarization dataset, enriching our research with human preferences as documented by the study [37]. Our methodology here incorporates an SFT model meticulously fine-tuned on expert-composed summaries of forum posts [3], operating within the TRLX framework for Reinforcement Learning from Human Feedback (RLHF), as introduced by Havrilla et al. [18]. The human preference dataset was gathered by Stiennon et al. [37] on samples from a different, but similarly-trained, SFT model.

## A.2 Mixture of *low gap* and *high gap*

In our previous discussion, we identified a *low gap* dataset, constituted by pairs of responses generated from the HH dataset. Given that these responses originate from the same model, we can infer that they represent semantically similar answers, hence the designation *low gap*. Concurrently, we maintain the positive samples constant while selecting negative samples generated by a Pythia 2.8B model. The significant performance disparity between the models results in a considerable variation in the quality of these negative samples, which we refer to as the *high gap*. We mix these two types of data in varying proportions to mimic the distribution of data in real-world scenarios. We label this approach the "mixture experiment." To facilitate a better comparison of the distributions across different ratios, we proceed to illustrate our findings with the following graph:

Figure 6 clearly illustrates that as the mixture ratio increases—that is, the proportion of high gap data rises—the dispersion of the overall dataset broadens. Conversely, a decrease in the mixture ratio, corresponding to an elevated presence of low gap data, results in a more concentrated distribution of reward discrepancy.

---

[3] https://huggingface.co/CarperAI/openai_summarize_tldr_sft

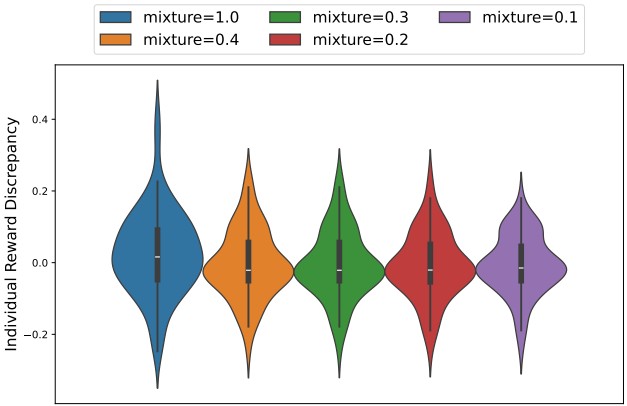

Figure 6: Distribution of individual reward discrepancies following the Pythia-2.8B model.

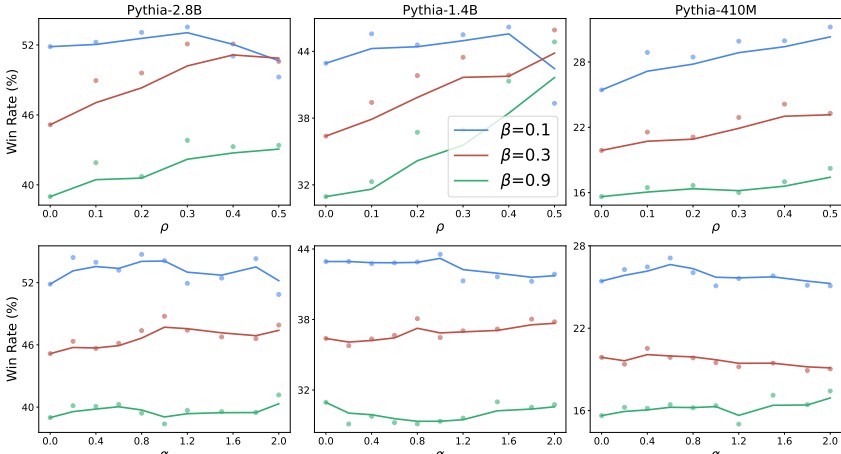

Figure 7: Performance comparison across different $\beta$ values and $\rho$ values for three different model sizes (Pythia-2.8B, Pythia-1.4B, and Pythia-410M) on the Anthropic HH dataset using GPT-4 as the evaluator. Each subplot represents the win rate for varying parameters $\beta = 0.1$, $0.3$, and $0.9$ with exponential smoothing.

## A.3 Hyperparameter Sensitivity

In this section, we investigate the impact of key hyperparameters in the $\beta$-DPO methodology, using the Anthropic HH dataset and leveraging models such as Pythia-2.8B and GPT-4 for evaluation. Specifically, we examine the influence of varying the hyperparameters $\alpha$ and the filtering ratio $\rho$. The parameter $\alpha$ is explored across a broad spectrum of values including $0.2$, $0.4$, $0.6$, $0.8$, $1.0$, $1.2$, $1.5$, $1.8$, and $2.0$. Concurrently, the filtering ratio $\rho$ is investigated at intervals ranging from $0.1$ to $0.5$, at a granularity of $0.1$. This comprehensive analysis aims to unravel how these hyperparameters affect the performance and outcomes of the $\beta$-DPO process.

**Static $\beta$ vs. Dynamic $\beta$.** Our results, as depicted in Figure 7, reveal that a dynamic $\beta$ (where $\alpha \neq 0$) prevails over a static $\beta$ (where $\alpha = 0$) under an exhaustive range of hyperparameter configurations. This outcome aligns seamlessly with our underlying premise: static beta fails to fully leverage a model's potential when confronted with varying data quality within a dataset. Furthermore, our observations highlight an intriguing trend: as $\alpha$ increases, the performance of the $\beta$-DPO model initially improves before declining, typically peaking within the interval of $0.6$ to $1.0$.

**Filtering Ratio $\rho$ Sensitivity.** As illustrated in the figure 7, varying model sizes exhibit distinct sensitivities to the parameter $\rho$. Within the context of the Pythia-2.8B model, a $\rho$ value of $0.3$ yields optimal performance, whereas for the Pythia-410M model, a $\rho$ value of $0.5$ is superior. This can be posited to suggest that smaller models may require more stringent data filtering to perform optimally, whereas larger models possess the increased capacity necessary for learning the most

effective strategies. This insight provides a significant directive for future research: the value of $\rho$ should be fine-tuned according to the specific circumstances of the application.

## A.4  The ablation study *w.r.t.* $M_0$

In this work, we employed a moving average updating scheme [24] for the updating of $M_0$. In order to investigate the superiority of this configuration, we also conducted a comparative experiment involving hyperparameter settings. Specifically, $M_0$ was treated as a constant in the training process, while all other settings remained unchanged. The experimental results are as follows:

Table 4: Comparison of win rates across varying $M_0$ in $\beta$-DPO.

| $M_0$ | 0 | 1 | 3 | 5 | 7 | 10 | Moving Average |
|---|---|---|---|---|---|---|---|
| $\beta$-DPO | 53.35 | 54.00 | 53.85 | 55.61 | 53.19 | 56.75 | 57.07 |

As demonstrated in Table 4, by tuning $M_0$, we were able to achieve significant performance improvements, which approached the performance level of the moving average updating scheme. This clearly underscores the superiority of the moving average updating scheme. On one hand, it obviates the need for an additional manual search process. On the other hand, it facilitates stable performance enhancements.

## A.5  Our Methods with Explicit Reward Model

To broaden the application of the $\beta$-DPO strategy for alignment and to address the limitations of DPO's implicit reward model, we incorporated two external reward models (RMs):

- We utilized llm-blender/PairRM [22] to score the chosen and rejected responses, denoted as $r_{w,\text{PairRM}}, r_{l,\text{PairRM}}$.
- We adopted the v0.2 Llama3-Instruct setting [26] by employing RLHFlow/ArmoRM-Llama3-8B-v0.1 [41] as the reward model to rank responses, denoted as $r_{w,\text{ArmoRM}}, r_{l,\text{ArmoRM}}$.

The other algorithmic processes (refer to Appendix Algorithm 1) remain unchanged, with modifications made only to $M_{i,\text{PairRM}} = r_{w,\text{PairRM}}^{(i)} - r_{l,\text{PairRM}}^{(i)}$ and $M_{i,\text{ArmoRM}} = r_{w,\text{ArmoRM}}^{(i)} - r_{l,\text{ArmoRM}}^{(i)}$. For comparison with baselines, we assess our models using one of the most popular open-ended instruction-following benchmarks: AlpacaEval 2. The detailed results are as follows:

Table 5: Comparison of different methods on Llama3-Instruct (8B) with explicit reward model

| Method | Llama3-Instruct (8B) | Llama3-Instruct (8B) |
|---|---|---|
| | LC (%) | WR (%) |
| DPO (Implicit RM) | 40.44 | 37.38 |
| $\beta$-DPO (Implicit RM) | **43.38** | **38.21** |
| SimPO (Implicit RM) | 44.38 | 38.97 |
| $\beta$-SimPO (Implicit RM) | **46.03** | **40.18** |
| SimPO (PairRM) | 44.70 | 38.98 |
| $\beta$-SimPO (PairRM, Instance-Level) | 43.84 | 38.54 |
| $\beta$-SimPO (PairRM, Batch-Level) | **45.65** | **39.76** |
| SimPO (ArmoRM) | 53.70 | 47.50 |
| $\beta$-SimPO (ArmoRM, Instance-Level) | 49.05 | 45.47 |
| $\beta$-SimPO (ArmoRM, Batch-Level) | **54.86** | **49.66** |

From the results above, we observed the following:

**The proposed dynamic $\beta$ strategy demonstrates strong generalizability.** Both $\beta$-D,SimPO consistently yield stable performance improvements across implicit and explicit RMs.

**Batch-level performance is crucial.** In explicit RM settings, batch-level dynamic $\beta$ consistently outperforms instance-level dynamic $\beta$.

### A.6 GPT-4 prompts for computing summarization and dialogue win rates

A fundamental element of our experimental framework involves the assessment of win rates facilitated by GPT-4. In this segment, we delineate the prompts employed to ascertain win rates pertinent to our summarization and dialogue-oriented investigations. For the entirety of our experimental endeavors, we utilize GPT-4. It is important to note that the sequence in which summaries or responses are presented is randomized for each evaluation.

**Summarization GPT-4 win rate prompt.**

```
Which of the following summaries does a better job of summarizing the most \
important points in the given forum post?

Post:
<post>

Summary A:
<Summary A>

Summary B:
<Summary B>

FIRST provide a one-sentence comparison of the two summaries, explaining which \
you prefer and why. SECOND, on a new line, state only "A" or "B" to indicate your \
choice. Your response should use the format:
Comparison: <one-sentence comparison and explanation>
Preferred: <"A" or "B">
```

**Dialogue GPT-4 win rate prompt.**

```
For the following query to a chatbot, which response is more helpful?

Query: <the user query>

Response A:
<either the test method or baseline>

Response B:
<the other response>

FIRST provide a one-sentence comparison of the two responses and explain \
which you feel is more helpful. SECOND, on a new line, state only "A" or \
"B" to indicate which response is more helpful. Your response should use \
the format:
Comparison: <one-sentence comparison and explanation>
More helpful: <"A" or "B">
```

## B  Broader Impacts

This paper presents work whose goal is to advance the field of Machine Learning. There are many potential societal consequences of our work, none of which we feel must be specifically highlighted here.

