# OpenReview forum: "$\beta$-DPO: Direct Preference Optimization with Dynamic $\beta$"
_NeurIPS.cc/2024/Conference — NeurIPS 2024 poster_

### Official Review · Reviewer_4RgK · 2024-06-12

**Soundness:** 1
**Presentation:** 4
**Contribution:** 2
**Rating:** 3
**Confidence:** 5

**Summary:**

This paper proposes a dynamic method to tune the hyperparameter $\beta$ in DPO according to the data in each batch. The proposed method is tested on Pythia-410M, 1.4B, and 2.8B base models. Compared to vanilla DPO, it performs better on Anthropic HH and Reddit TL;DR summarization datasets.

**Strengths:**

1. The topic is meaningful. The method of tuning hyperparameter $\beta$ potentially influences the RLHF or alignment research if the work is solid.

2. The presentation is excellent. The paper is easy to read.

**Weaknesses:**

The proposed method is not sound, neither in theory nor experiments.

1. The experiences are only based on small models, Pythia-410M, 1.4B, and 2.8B. It is hard to evaluate the performances on SOTA (relatively large) models, e.g., 7B or 8B models.

2. In theory, this paper fails to show how the dynamic terms are derived. It gives the dynamic terms directly with more hyperparameters.

3. In practice, the vanilla DPO only has one hyperparameter $\beta$. This paper aims to solve the hyperparameter tuning problem. However, it solves this problem by introducing two hyperparameters $\beta_{0}$ and $\alpha$. With more hyperparameters and space to tune, it is not surprising to fit better, especially on small models.

**Questions:**

1. $\beta$ is not a simple learning rate. $\beta$ is derived from the KL-divergence penalty term. It ensures that the tuned policy is not "far away" from the original policy. Thus, it should be based on global information, which is the whole training data in offline RLHF. However, in this paper, the dynamic $\beta$ is only based on batch-level information. How does the algorithm capture the global information through batch-level data?

2. The proposed algorithm has two hyperparameters $\beta_{0}$ and $\alpha$. Is it harder to tune compared to the vanilla DPO with only one hyperparameter $\beta$?

**Limitations:**

The authors addressed the limitations of the paper. Here are some additions.

1. The proposed method is only tested on small models, Pythia-410M, 1.4B, and 2.8B. It is hard to evaluate the performances on SOTA (relatively large) models, e.g., 7B or 8B models.

2. The winning rates are only evaluated by GPT-4 instead of real human groups. But this is a common issue in most LLM studies, and may be acceptable for research.

---

> ### Author Rebuttal · Authors · 2024-08-07
>
> Dear Reviewer,
>
> Thanks for your kind review. We are glad that you found our paper meaningful and easy to follow. We provide detailed answers to your comments below.
>
> **Q1: It is hard to evaluate the performances on SOTA (relatively large) models, e.g., 7B or 8B models.**
> > A1: Thank you for raising this concern. To expand our approach to more diverse datasets and model sizes, we follow the current state-of-the-art models SimPO[3]. We perform $\beta$-DPO with two families of models, Llama3-8B-Instruct and Mistral-7B-Instruct, on UltraChat-200k UltraFeedback. For comparison with baselines, we assess our models using one of the most popular open-ended instruction-following benchmarks: AlpacaEval 2. All settings are consistent with SimPO [1].
> Please refer to Table below:
>
> | Method | Mistral-Instruct (7B) | Mistral-Instruct (7B) | Llama3-Instruct (8B) | Llama3-Instruct (8B) |
> |--------|----------------------|-----------------------|---------------------|---------------------|
> |        | LC (%)               | WR (%)                | LC (%)              | WR (%)              |
> | DPO    | 20.98                | **21.60**             | 40.44               | 37.38               |
> | $\beta$-DPO | **23.56**       | 20.42                 | **43.38**           | **38.21**           |
> | SimPO  | 28.50                | 30.56                 | 44.38               | 38.97               |
> | $\beta$-SimPO | **30.48**     | **32.13**             | **46.03**           | **40.18**           |
>
> > Table: AlpacaEval 2 results under the Mistral-Instruct (7B) and Llama3-Instruct (8B). LC and WR denote length-controlled and raw win rate, respectively. Regardless of whether we use Llama3-8B or Mistral-7B, and whether the loss function is DPO or SimPO, our $\beta$-{D, Sim}PO strategy consistently demonstrates significant performance improvements. This thoroughly showcases the method's strong generalization ability and excellent scalability.
>
> **Q2: This paper fails to show how the dynamic terms are derived.**
> > A2: Thank you for your suggestion. Our work primarily identifies an empirical relationship between $\beta$ and data quality, validated across various datasets and model architectures. While we recognize that the theoretical foundations warrant further exploration, the proposed dynamic strategy is nonetheless notable, providing a new and effective paradigm for fine-tuning large models and studying data quality.
>
> **Q3: Is it harder to tune compared to the vanilla DPO with only one hyperparameter?**
> > A3: Our proposed $\beta$-DPO is a straightforward, efficient, and easily transferable strategy. Firstly, the selection of $\beta\_0$ in all instances of $\beta$-DPO is consistent with the $\beta$ used in DPO, with a default choice of 0.1 (0.01 for UltraChat-200k), eliminating the need for extensive hyperparameter tuning for $\beta$. As for the uniquely introduced $\alpha$, we find:
>
> > **In most scenarios, setting $\alpha = \frac{2}{M_0}$ yields stable performance improvements, where $M_0$ can be estimated using a moving average updating scheme (refer to Equation 7).** This is informed by the formula $\beta\_{\text{batch}} = [1 + \alpha(\mathbb{E}\_{i \sim \text{batch}}[M\_i] - M\_0)]\beta\_0$, resulting in an overall change range of $[\frac{2\mathbb{E}\_{i \sim \text{batch}}[M\_i] - M\_0}{M\_0}]\beta\_0$, which normalizes based on $M\_0$ over the foundation of $\beta\_0$.
>
> | | HH | TLDR |
> |-------------------|-------------------|-------------|
> | DPO               | 51.01               |   32.45       |
> | $\beta$-DPO               | 57.68              | 51.67|
> | $\beta$-DPO ($\frac{2}{M\_0}$)              | 58.02               | 51.32 |
>
> > To substantiate this perspective, we present performance in the above table, demonstrating that our setting achieves significant enhancements across various datasets and models compared to DPO, **without imposing additional pressure for hyperparameter searches.** We appreciate your concern; while we believe that further theoretical consolidation is a meaningful future endeavor, we maintain that the $\beta$-DPO approach remains valuable, offering a straightforward (not overly reliant on hyperparameter tuning) and effective (stable performance enhancements) new paradigm for fine-tuning large models and studying data quality.
>
> **Q4: How does the algorithm capture the global information through batch-level data?**
> > A4: We employed a moving average estimator to estimate the global reward margin, as referenced in Equation 7. This technique has been commonly used in deep learning [1, 2].
>
> **Q5: The winning rates are only evaluated by GPT-4 instead of real human groups.**
> > A5: Thank you for your suggestion. As it stands, the current research landscape indicates that GPT-4 provides the most common strategy for validation. Moreover, the alignment between GPT-4 assessments and human evaluations has been sufficiently established in existing benchmarks (AlpacaEval 2).
>
> [1] Yuan et. al. Provable stochastic optimization for global contrastive learning: Small batch does not harm performance. ICML2022.
>
> [2] ADAM: A METHOD FOR STOCHASTIC OPTIMIZATION. ICLR 2015.
>
> [3] SimPO: Simple Preference Optimization with a Reference-Free Reward. Yu Meng, Mengzhou Xia, and Danqi Chen.

---

> > ### Comment · Reviewer_4RgK · 2024-08-13
> >
> > Thank the authors for the response.
> >
> > 1. As the calibration is better at the batch level than the instance level, do you have an ablation on batch size $b$? (Sorry to bring up this concern late.)
> >
> > 2. Is it equivalent to calibrating the learning rate instead of $\beta$?

---

> > > ### Author Response · Authors · 2024-08-14
> > > **Follow-up Discussion**
> > >
> > > Thank you for your valuable comments and suggestions on our submission. Your suggestions to **1) assess the performance on SOTA models with relatively large scale, such as 7B or 8B models; 2) compare the performance with the vanilla DPO using only one hyperparameter through automated parameter tuning; 3) conduct an ablation study on batch size; and 4) explore the calibration of learning rate and dynamic $\beta$** have significantly contributed to enhancing the coherence and impact of our work. We hope that these improvements will be taken into account during the evaluation process.
> > >
> > > If our response has resolved your concerns on our paper, we will greatly appreciate it if you could re-evaluate our paper. Should you have any further questions or need additional clarification, please know that we are eager and prepared to continue our discussions.

---

> > > > ### Author Response · Authors · 2024-08-14
> > > > **Inquiry on Additional Feedback**
> > > >
> > > > Thanks for your constructive feedback on our paper. We kindly inquire whether there may exist any additional concerns or unresolved questions that might be impeding the paper's attainment of a higher rating. We are available for any further clarifications or discussions!

---

> ### Author Response · Authors · 2024-08-13
>
> Thank you for your response. Regarding the question about batch size and calibrating the learning rate, we provide the following elaboration:
>
> **Q1: As the calibration is better at the batch level than the instance level, do you have an ablation on batch size?**
>
> > **A1:** We have experimented with different batch sizes for $\beta\_{\text{Batch}}$ on the Pythia-410M model, and the results are as follows:
> > | Batch Size | WR (%) |
> > |------------|--------|
> > | 128        | 30.18  |
> > | 64         | 29.35  |
> > | 32         | 28.78  |
>
> > The above results demonstrate that larger batch sizes lead to better model performance on Pythia-410M model. Our intuitive understanding of this observation is:
> >  - Larger batch sizes enable more accurate estimation of $\beta$.
> >  - In the extreme case where batch size = 1, batch-level calibration degrades to instance-level, resulting in high instability in the estimation of $\beta\_{\text{Batch}}$.
> >
> >  We appreciate your suggestion. Due to the limited time for discussion, we will further investigate this conclusion's reliability by experimenting with other model sizes (Pythia 2.8B, Mistral-Instruct (7B) | Llama3-Instruct (8B)).
>
> **Q2: Is it equivalent to calibrating the learning rate instead of $\beta$?**
>
> > **A2:** Dynamic $\beta$ is not equivalent to calibrating the learning rate. We can further analyze this issue from the perspective of gradients.
> > $$\nabla\_\theta \mathcal{L}\_\text{DPO}(\pi\_\theta;\pi\_{\text{ref}}) = -\beta\mathbb{E}\_{(x, y\_w, y\_l) \sim \mathcal{O}}[\sigma(\hat{r}\_\theta(x, y\_l) - \hat{r}\_\theta (x, y\_w))  (\nabla\_{\theta, y\_w} - \nabla\_{\theta, y\_l})]$$
> > where $\sigma(\hat{r}\_\theta(x, y\_l) - \hat{r}\_\theta (x, y\_w)) =\sigma(\beta \log \frac{\pi\_{\theta}(y\_l|x)}{\pi\_{\text{ref}}(y\_l|x)} - \beta \log \frac{\pi\_{\theta}(y\_w|x)}{\pi\_{\text{ref}}(y\_w|x)})$.
>
> > Therefore, the gradients are correlated with $\beta \sigma(\beta [\log \frac{\pi\_{\theta}(y\_l|x)}{\pi\_{\text{ref}}(y\_l|x)} - \log \frac{\pi\_{\theta}(y\_w|x)}{\pi\_{\text{ref}}(y\_w|x)}] )$, **but non-linearly correlated with $\beta$**. Here, $[\log \frac{\pi\_{\theta}(y\_l|x)}{\pi\_{\text{ref}}(y\_l|x)} - \log \frac{\pi\_{\theta}(y\_w|x)}{\pi\_{\text{ref}}(y\_w|x)}] < 0$. Consequently, as $\beta$ increases, $\sigma(\beta [\log \frac{\pi\_{\theta}(y\_l|x)}{\pi\_{\text{ref}}(y\_l|x)} - \log \frac{\pi\_{\theta}(y\_w|x)}{\pi\_{\text{ref}}(y\_w|x)}] )$ approaches 0, and the entire gradient also approaches 0.
>
> > Considering the gradient update:
> `params = old_params - lr * grad`
> Directly calibrating the learning rate cannot achieve the same effect as calibrating $\beta$. However, if an appropriate construction of `lr` can be found (high gap --> small learning rate), the experience from $\beta$-DPO suggests that there may be some improvements.
>
> We sincerely appreciate your valuable feedback and eagerly anticipate integrating these improvements into our manuscript. Please let us know if you have any further concerns, and we are encouraged to have a discussion.

---

### Official Review · Reviewer_uGzu · 2024-06-15

**Soundness:** 3
**Presentation:** 3
**Contribution:** 3
**Rating:** 6
**Confidence:** 3

**Summary:**

This paper studies the relation between the best $beta$ parameter of DPO and the data quality. Motivated from the observation, the authors propose a way to dynamically choose the $beta$ at the batch level. The evaluation shows the proposed method improves DPO and its variants' (IPO, KTO, SPPO) performance on Antrhopic-HH dataset.

**Strengths:**

* The proposed method is motivated from empirical finding that the optimal beta tends to be positively correlated with the reward discrepancy.
* The proposed method is based on highly principled ideas such as batch average, exponential moving average, gaussian pdf.
* Experiments demonstrate the effectiveness of the proposed method over DPO, IPO, KTO, SPPO

**Weaknesses:**

* The experiment mainly relies on Anthropic HH dataset, containing 170000 dialogues. It would be nice if the author can verify the results on another dataset with different domains.

**Questions:**

* $\beta_{batch}$ is defined in Eq. (6). What is a usual range of $\beta_{batch}$ in experiment? Is it possible that $\beta_{batch}$ goes to negative?
* In section 4.2.2 data filtering, the authors adopt a sampling without replacement with weights defined in Eq. (8). This cannot completely remove the outlier (i.e., there are still small chance for the outliers to be selected). What if you sort the data in an increasing order according to $|M_i-M_0|$ and select the top $|data|\times \rho$ instances? (this can completely remove the outliers) How do you compare your method with this?

**Limitations:**

No potential negative societal impact.

---

> ### Author Rebuttal · Authors · 2024-08-07
>
> Dear Reviewer,
>
> Thanks for your kind review. We are glad that you found our paper meaningful and easy to follow. We provide detailed answers to your comments below.
>
> **Q1: It would be nice if the author can verify the results on another dataset with different domains.**
> > A1: Thank you for raising this concern. To expand our approach to more diverse datasets and model sizes, we follow the current state-of-the-art models SimPO[1]. We perform $\beta$-DPO with two families of models, Llama3-8B-Instruct and Mistral-7B-Instruct, on UltraChat-200k UltraFeedback. For comparison with baselines, we assess our models using one of the most popular open-ended instruction-following benchmarks: AlpacaEval 2. All settings are consistent with SimPO [1].
> Please refer to Table below:
>
>
> | Method | Mistral-Instruct (7B) | Mistral-Instruct (7B) | Llama3-Instruct (8B) | Llama3-Instruct (8B) |
> |--------|----------------------|-----------------------|---------------------|---------------------|
> |        | LC (%)               | WR (%)                | LC (%)              | WR (%)              |
> | DPO    | 20.98                | **21.60**             | 40.44               | 37.38               |
> | $\beta$-DPO | **23.56**       | 20.42                 | **43.38**           | **38.21**           |
> | SimPO  | 28.50                | 30.56                 | 44.38               | 38.97               |
> | $\beta$-SimPO | **30.48**     | **32.13**             | **46.03**           | **40.18**           |
>
> > Table: AlpacaEval 2 results under the Mistral-Instruct (7B) and Llama3-Instruct (8B). LC and WR denote length-controlled and raw win rate, respectively. Regardless of whether we use Llama3-8B or Mistral-7B, and whether the loss function is DPO or SimPO, our $\beta$-{D, Sim}PO strategy consistently demonstrates significant performance improvements. This thoroughly showcases the method's strong generalization ability and excellent scalability.
>
> **Q2: What is a usual range of beta in experiment? Is it possible that goes to negative?**
>
> > A2: Our initial $\beta\_0$ is set to 0.1, and the range of beta in experiments falls within [0.0, 0.4]. The specific range of variation is detailed in `REBUTTAL Figure 2 Middle`.
> Experimentally, $\beta\_{\text{batch}}$ can become negative. According to Equation 6, a negative value implies that the reward discrepancy is negative or $\mathbb{E}\_{i \sim \text{batch}}[M\_i]$ is significantly lower than the global mean of $M\_i$, indicating a high likelihood of outliers. Therefore, we apply a cutoff at 0 to ensure training stability, $\beta\_{\text{batch}} = \max(0.0, \beta\_{\text{batch}})$.
>
> **Q3: What if you sort the data in an increasing order and select top instances?**
> > A3: Thank you for pointing this out. This method represents a hard selection approach, whereas our method utilizes soft selection. The hard selection concept aligns with what we have shown in Figure 5 Left (Filter Head & Tail), and we believe both are effective filtering methods. Both approaches validate that extreme $M\_i$ values are likely outliers that need filtering.
>
> > **It is important to highlight that this work does not propose a novel filtering method, but we find that filtering enhances stability.** As shown in Figure 5 Left, dynamic scheduling could also improve other filtering methods. We are confident that our dynamic schedule will continue to provide stable performance improvements as better LLM data filtering techniques emerge in the future.
>
> [1] SimPO: Simple Preference Optimization with a Reference-Free Reward. Yu Meng, Mengzhou Xia, and Danqi Chen.

---

> > ### Comment · Reviewer_uGzu · 2024-08-11
> > **Post-rebuttal update**
> >
> > This is to confirm that I have reviewed the author's response and the materials given by the authors during the rebuttal. These responses have answered my questions. However, after reading other reviewers' questions, I feel there are some other works to do. So I decided to keep my initial rating.

---

### Official Review · Reviewer_UzDb · 2024-07-12

**Soundness:** 3
**Presentation:** 2
**Contribution:** 2
**Rating:** 6
**Confidence:** 3

**Summary:**

This work proposes two techniques to improve the performance of the popular DPO alignment algorithm. The first technique proposes a strategy to dynamically adapt the $\beta$ coefficient of DPO, which controls the strength of the KL penalty with respect to the reference policy. The second technique proposes a filtering mechanism that uses reward margins (the difference between the rewards of the preferred and dispreferred response) to filter the data being sampled for DPO training. Results show that the two techniques complement each other and lead to increased win rates in a range of settings.

**Strengths:**

1.	The paper is well-organized and technically sound. The general flow of the paper is smooth. The paper has an appropriate number of citations and properly details existing work in the related work section.
2.	The method presented is simple to understand and easy to integrate into any DPO implementation.
3.	The results are promising and are giving consistent improvements in a range of scenarios.

**Weaknesses:**

1) Technical and experimental details need to be clearly clarified (refer to questions). Currently, I am unable to see how the techniques presented are related, they seem to be independent.
2) Writing can be improved. In particular, there are very few details about the reward models used in the paper and no acknowledgment of the fact that in contrast to vanilla DPO, the proposed method is reliant on access to a reward model.

**Questions:**

1.	The paper makes an implicit assumption that low-gap examples are high-quality. However, I do not think this is generally true. For example, if both the preferred and dispreferred response are rated very poorly by the reward model, they will also have low-gap. Having clearer writing in this regard and a general explanation of this would be useful.
2.	On what criteria does the paper make the claim that Anthropic HH is a low-gap dataset? In my experience, the dataset contains numerous pairs that are easily separable as well. Is any dataset where both $y_w$ and $y_l$ are sampled from the same policy considered low-gap?
3.	In contrast to DPO, this method necessitates learning/using an off-the-shelf reward model. There are very few details about this reward model in the paper. Currently, the paper assumes that the reward gaps are generally available, however this is often not the case. It would also be good to have a few sentences acknowledging this difference between DPO and the proposed method.
4.	How is beta being used in data filtering? $\beta$ is not mentioned anywhere in Section 4.2.2. The data filter seems like a normal reward margin-based filter.
5.	How is the selective filtering ablation run? What does arranging the gradients mean? Is gradient referring to the reward margin?
6.	Is m (the momentum coefficient) also a hyper parameter? The claim of the paper is that $\alpha$ is the only hyperparameter.
7.	Current results are only on the Pythia class of models and it would be interesting to see the same improvements on a different class of models. I think that is necessary to substantiate the claim of model agnosticity.

---

> ### Author Rebuttal · Authors · 2024-08-07
>
> Dear Reviewer,
>
> Thanks for your kind review. We are glad that you found our paper meaningful and easy to follow. We provide detailed answers to your comments below.
>
> **Q1: There are very few details about the reward models used in the paper**
> > A1: Thank you for pointing this out. **We directly use the implicit reward model induced by the policy trained by DPO, where the reward discrepancy in DPO is expressed as: $ \beta \log (\frac{\pi\_\theta (y\_w \mid x) }{\pi\_{\text{ref}}(y\_w \mid x)}) - \beta \log (\frac{\pi\_\theta (y\_l \mid x) }{\pi\_{\text{ref}}(y\_l \mid x)})$.**
>
>
> **Q2: Why are low-gap examples considered high-quality?**
> > A2: We appreciate your concern regarding this matter. First, it is crucial to clarify that both the low-gap and high-gap examples are derived from data of a certain quality, rather than from extremely poor quality sources characterized by meaningless chatter.
> **We posit that datasets of extremely poor quality would not be utilized for training at this stage.** Furthermore, we can categorize the cases of $y\_w$ and $y\_l$ into the following four types:
>
> | Quality of $y\_w$ | Quality of $y\_l$ | Description | Behavior of $\beta$-DPO |
> |-------------------|-------------------|-------------|-------------------------|
> | Low               | Low               |   High-quality, hard discriminated pairs       | Lower $\beta$ → Assertive updates |
> | Low               | High              | Label flipping, noise | Larger $\beta$ → Cautious updates |
> | High              | Low               | Easily discriminated pairs | Larger $\beta$ → Cautious updates |
> | High              | High              | High-quality, closely matched pairs | Lower $\beta$ → Assertive updates |
>
> > To further elucidate, both preferred and dispreferred responses are low-quality, yet they serve a beneficial purpose. We follow the settings of controlled sentiment generation in DPO. With access to the ground-truth reward function (a sentiment classifier), we control the quality of $y\_w$ and $y\_l$, where high-quality response is derived from a fine-tuned GPT-2 large model and low-quality response from an unrefined GPT-2 large model. The following table, derived from the IMDB test dataset, illustrates the KL-divergence and the associated rewards:
>
> |   KL-divergence     | 2  | 4  | 6  | 8  | 10  | 12  | 14 |
> |---------------------|----|----|----|----|----|----|----|
> | Both High-quality   | 65.68 | 89.16 | 94.26 | 96.24 | 97.74 | 98.91 | 98.58 |
> | Both Low-quality    | 63.72 | 80.42 | 84.00 | 85.00 | 81.80 | 82.20 | 81.75 |
> | High-quality $y\_w$, Low-quality $y\_l$ | 45.93 | 39.55 | 39.46 | 36.59 | 29.98 | 30.71 | 31.27 |
> > The results indicate that low-quality responses can be even more meaningful for model improvement than high-gap examples.
>
> **Q3: Why is the Anthropic HH dataset considered low-gap? Are there better datasets available?**
> > A3: The terms "low-gap" and "high-gap" are employed in a relative context. In comparison to the negative samples generated by SFT, the overall distribution of Anthropic HH exhibits smaller differences. To support this assertion, please refer to Appendix Figure 6, where positive samples originate solely from the HH dataset, while negative samples consist of a mix from SFT-generated and original negative samples from Anthropic HH. As the proportion of original Anthropic HH samples increases (i.e., the mixture ratio decreases), the distribution of reward margins becomes more concentrated.
>
> **Q4: How is beta utilized in data filtering?**
> > A4: In this work, the reward margin informs both the choice of $\beta$ and the framework for data filtering, establishing a bridge between $\beta$ and data filtering. The specific expression is:
> $p(\beta\_i)= \frac{1}{\sqrt{2\pi}\sigma}\exp(-\frac{(\beta\_i/\beta\_0-1)^2}{2\sigma^2\alpha^2})$
>
> **Q5: How is the selective filtering ablation conducted?**
> > A5: We apologize for any misunderstanding in our expression. Your interpretation is correct; we operate based on the reward margin. We will revise this section for clarity. In the DPO formulation, the sample gradients are strictly negatively correlated with the reward margin, implying that a larger reward margin corresponds to a smaller gradient. Therefore, during loss computation for each batch, we sort samples based on their reward margins and conduct selection accordingly. In the experiment corresponding to Figure 5 (Left), "Filter Tail 20%" refers to the filtering of the 20% of samples with the largest gradients, which corresponds to those with the smallest 20% reward margins, and vice versa.
>
> **Q6: Is m (the momentum coefficient) also a hyperparameter?**
> > A6: m (the momentum coefficient) is indeed a hyperparameter. However, in all our experiments, we consistently set it to **0.9 without any hyperparameter search**. To further illustrate, we present additional comparisons in `REBUTTAL Table 2`, showing that this parameter has a minimal effect on model performance.
>
> **Q7: Current results are only on the Pythia class of models**
> > A7: Thank you for raising this concern. To expand our approach to more diverse datasets and model sizes, we follow the current state-of-the-art models [1]. We perform beta-DPO with two families of models, Llama3-8B-Instruct and Mistral-7B-Instruct, on UltraChat-200k UltraFeedback. For baseline comparisons, we assess our models using one of the most popular open-ended instruction-following benchmarks: AlpacaEval 2. All settings are consistent with SimPO [1].
> > Please refer to `REBUTTAL Table 1`. Regardless of whether we use Llama3-8B or Mistral-7B, and whether the loss function is DPO or SimPO, our $\beta$-{D, Sim}PO strategy consistently demonstrates significant performance improvements. This thoroughly showcases the method's strong generalization ability and excellent scalability.
>
> [1] SimPO: Simple Preference Optimization with a Reference-Free Reward. Yu Meng and Mengzhou Xia and Danqi Chen.

---

> > ### Comment · Reviewer_UzDb · 2024-08-10
> > **Reply to rebuttal**
> >
> > I thank the authors for the efforts they have put in the paper and rebuttal. When writing my original review, I was almost certain that an external reward model is being used. The paper never mentioned the use of DPO as an implicit reward model and was generally unclear on this very important detail. Section 3 has no mention of DPO as an implicit reward model and instead introduces the standard external RM, which led me to believe that an external RM is being used. This ambiguity was the main concern of reviewer fXTA as well.
> > Upon learning that the model is in fact an implicit reward model, I believe that many more clarification/experiments are necessary -
> > 1. How does this method perform with an external reward model?
> > 2. I think the method presented is preventing overfitting using margin-based regularization. The open-source community has noted that margins in DPO explode if it is trained for multiple epochs. By increasing the KL penalty for higher margins, this method might be regularizing against such overfitting. This is definitely worth exploring more.
> > 3. At the start of training, all margins should theoretically be $0$ as $\pi = \pi_{ref}$. One of the implications of this is that data points sampled early during training will increase in margin (as they are being trained on). I am unsure of the implications but the data ordering introduces some bias.
> > 4. The non-stationarity is particularly concerning. Moreover, the intermediate states during DPO training all correspond to different reward models. The reward margin for any given datapoint is thus changing at each tilmestep. I think more analysis of how these margin trajectories look like for different datapoints will be very interesting and improve the quality of the paper.
> >
> > I do note that the results for the current method are promising, but I believe that this paper has potential to be a much better one. I will maintain my current score for now, but will follow the discussion and possibly revise based on more discussion with reviewers and authors.

---

> > > ### Author Response · Authors · 2024-08-10
> > >
> > > Thank you for your response. We apologize for not explicitly mentioning that the model is an implicit reward model in the initial draft. We will promptly revise this in the next version. Regarding your question about implicit rewards, we respond as follows:
> > >
> > > **1. How does this method perform with an external reward model?**
> > > > Thank you for highlighting this point. Computing the external reward for the training set of 160k samples would 1) significantly increase computational complexity and 2) make the reward highly sensitive to the external reward model used for annotation. Such an approach is difficult to generalize to arbitrary data scenarios and DPO-like methods. Previous studies have raised concerns about the computational complexity to external reward models [1], and it has been suggested that employing implicit rewards could potentially lead to better alignment [2].
> > > We appreciate your suggestion and agree that this is indeed an aspect that could be refined.
> > >
> > > **2. I think the method presented is preventing overfitting using margin-based regularization.**
> > > > Thank you for your suggestion. We believe this is one of the advantages and motivations behind $\beta$-DPO. As stated in lines 151-152 of the original text: "Conversely, for high gap pairwise data, maintaining a low $\beta$ may lead to overfitting, which significantly undermines the alignment process."
> > >
> > > **3. I am unsure of the implications but the data ordering introduces some bias.**
> > > > We suggest that data ordering has minimal impact. Please refer to `REBUTTAL Figure 2 Middle` (The value of $\beta_{\text{Batch}}$ and preference accuracy along the training steps.). The overall distribution of $\beta_{\text{Batch}}$ does not tend to disperse, indicating that the corresponding margin distribution is relatively stable. In the early stages of training (< 10k / 160k steps), the performance of $\beta$-DPO is similar to DPO; as the model's discriminative power improves, $\beta_{\text{Batch}}$ varies dynamically within the range of [0, 0.4] without further continuous amplification of its value range.
> > >
> > > **4. The non-stationarity is particularly concerning. The reward margin for any given datapoint is thus changing at each timestep.**
> > > > The moving average updating scheme on $M_0$ (ref to Equation 7) helps mitigate the potential instability. Although the reward margin increases further ($M_i$ increases) as the model's capability improves, $\beta_{\text{Batch}}$ is positively correlated with $M_i - M_0$. As a result, $\beta$-DPO focuses more on the dynamic reward of different datapoints relative to global datapoints, rather than their absolute values. This approach may help reduce the impact of the changing reward margins on the overall stability of the system. The evolution of $\beta_{\text{Batch}}$ over the course of training steps, as illustrated in `REBUTTAL Figure 2 Middle`, corroborates this perspective.
> > >
> > > We sincerely appreciate your valuable feedback and eagerly anticipate integrating these improvements into our manuscript. Please let us know if you have any further concerns, and we are encouraged to have a discussion.
> > >
> > > [1] Filtered Direct Preference Optimization. https://arxiv.org/pdf/2404.13846.
> > >
> > > [2] Bootstrapping Language Models with DPO Implicit Rewards. https://arxiv.org/pdf/2406.09760

---

> > > > ### Comment · Reviewer_UzDb · 2024-08-11
> > > > **Reply**
> > > >
> > > > Thank you for the prompt response. Although I still believe that this paper has scope for improvement, I have raised my score because of a) promising results and b) the effort put in by the authors to make the rebuttal productive.
> > > >
> > > > I still believe that this method needs to be tested with an external RM to give further insights. Strong off-the-shelf reward models trained on diverse sources of data are now available openly, and thus it is not necessary to train a RM from scratch, only inference is required. DPO's implicit reward model is generally poorer than an explicit RM, this can also been seen on the RewardBench leaderboard [1]. Finally, I do not completely understand the sensitivity to reward point that the authors brought up, the same issues should exist with DPO as well (even more so as DPO training can often be unstable).
> > > >
> > > > [1]: Lambert, N., Pyatkin, V., Morrison, J., Miranda, L. J., Lin, B. Y., Chandu, K., ... & Hajishirzi, H. (2024). Rewardbench: Evaluating reward models for language modeling. arXiv preprint arXiv:2403.13787.

---

> > > > > ### Author Response · Authors · 2024-08-13
> > > > > **We sincerely appreciate each valuable comment and suggestion.**
> > > > >
> > > > > First, we would like to express our gratitude for your positive assessment of our work, particularly regarding a) the promising results and b) the authors' effort in making the rebuttal constructive. Regarding the question about external reward models (RM), we provide the following elaboration:
> > > > >
> > > > > **Q: I still believe that this method needs to be tested with an external RM to provide further insights.**
> > > > >
> > > > > > **A:** To broaden the application of the $\beta$-DPO strategy for alignment and to address the issue of DPO's poor implicit reward model, we sampled two external RMs:
> > > > > >
> > > > > > (1) We utilized llm-blender/PairRM [1] to score the chosen and rejected responses, denoted as $r\_{w,PairRM}, r\_{l,PairRM}$.
> > > > > >
> > > > > > (2) We adopted the v0.2 Llama3-Instruct setting [3] by employing RLHFlow/ArmoRMLlama3-8B-v0.1 [2] as the reward model to rank generated data, denoted as $r\_{w,ArmoRM}, r\_{l,ArmoRM}$.
> > > > > >
> > > > > > **The other algorithmic processes (refer to Appendix Algorithm 1) remain unchanged, with modifications made only to $M\_{i,\text{PairRM}}=r\_{w,\text{PairRM}}^{(i)} - r\_{l,\text{PairRM}}^{(i)}$ and $M\_{i,\text{ArmoRM}}=r\_{w,\text{ArmoRM}}^{(i)} - r\_{l,\text{ArmoRM}}^{(i)}$.** Due to time constraints, we only tested the implicit reward model on the state-of-the-art loss function (SimPO [3]). The detailed results are as follows:
> > > > >
> > > > > | Method  | Llama3-Instruct (8B) | Llama3-Instruct (8B) |
> > > > > |---------|-----------------------|-----------------------|
> > > > > |         | LC (%)                | WR (%)                |
> > > > > | DPO (implicit RM) | 40.44               | 37.38                 |
> > > > > | $\beta$-DPO (implicit RM) | **43.38**      | **38.21**            |
> > > > > |         |                       |                       |
> > > > > | SimPO (implicit RM) | 44.38               | 38.97                 |
> > > > > | $\beta$-SimPO (implicit RM)   | **46.03**     | **40.18**            |
> > > > > |         |                       |                       |
> > > > > |         |                       |                       |
> > > > > | SimPO (PairRM)   | 44.7                | 38.98                 |
> > > > > | $\beta$-SimPO (PairRM, Instance-Level)   | 43.84 | 38.54            |
> > > > > | $\beta$-SimPO (PairRM, Batch-Level)   | **45.65**  | **39.76**         |
> > > > > |         |                       |                       |
> > > > > |         |                       |                       |
> > > > > | SimPO (ArmoRM)   | 53.7                | 47.50                 |
> > > > > | $\beta$-SimPO (ArmoRM, Instance-Level)   | 49.05   | 45.47            |
> > > > > | $\beta$-SimPO (ArmoRM, Batch-Level)   | **54.86**  | **49.66**        |
> > > > >
> > > > > From the results above, we observed the following:
> > > > >
> > > > > 1. The proposed dynamic $\beta$ strategy demonstrates strong generalizability. Both $\beta$-{D,Sim}PO consistently yield stable performance improvements across implicit and explicit RMs.
> > > > > 2. Batch-level performance is crucial. In explicit RM, batch-level consistently outperforms instance-level dynamic $\beta$.
> > > > > 3. The algorithm is straightforward to implement without the need for additional parameter tuning. Due to time limitations for discussions, we conducted limited parameter tuning and found that setting $\alpha = \frac{2}{M\_0}$ yields stable performance improvements. This aligns with the formula $\beta\_{\text{batch}} = [1 + \alpha(\mathbb{E}\_{i \sim \text{batch}}[M\_i] - M\_0)]\beta\_0$, resulting in an overall change range of $[\frac{2\mathbb{E}\_{i \sim \text{batch}}[M\_i] - M\_0}{M\_0}]\beta\_0$, which normalizes using $M\_0$ based on $\beta\_0$. Similar experiments are also illustrated in [Rebuttal To Reiewer fXTA](https://openreview.net/forum?id=ZfBuhzE556&noteId=3L2HCyuMFP) and [Rebuttal To Reiewer 4RgK](https://openreview.net/forum?id=ZfBuhzE556&noteId=9JaGJcZ8w8).
> > > > >
> > > > > We sincerely appreciate your suggestions regarding our work, and we look forward to integrating these improvements into our manuscript. We are grateful for your contributions to refining $\beta$-DPO during this rebuttal phase!
> > > > >
> > > > > [1] Dongfu Jiang, Xiang Ren, and Bill Yuchen Lin. "LLM-Blender: Ensembling large language models with pairwise ranking and generative fusion." In ACL, 2023.
> > > > > [2] Haoxiang Wang, Wei Xiong, Tengyang Xie, Han Zhao, and Tong Zhang. "Interpretable preferences via multi-objective reward modeling and mixture-of-experts." arXiv preprint arXiv:2406.12845, 2024.
> > > > > [3] Yu Meng, Mengzhou Xia, and Danqi Chen. "SimPO: Simple Preference Optimization with a Reference-Free Reward." arXiv preprint arXiv:2405.14734, 2024.

---

> > > > > > ### Comment · Reviewer_UzDb · 2024-08-13
> > > > > > **Reply**
> > > > > >
> > > > > > Thank you! These results definitely further increase the strength of the paper, allowing you to strengthen the claim that the method is "reward-model independent". Although the win rate improvements are less significant than the improvements seen with implicit models, I also understand that the short turnover period means that these can probably be pushed up further with more tuning.
> > > > > >
> > > > > > I have raised my score further.

---

### Official Review · Reviewer_3526 · 2024-07-13

**Soundness:** 3
**Presentation:** 3
**Contribution:** 3
**Rating:** 6
**Confidence:** 4

**Summary:**

The paper presents an improvement in Direct Preference Optimization (DPO), a method for aligning and fine-tuning large language models (LLMs) based on human preferences. The authors identify two critical factors affecting DPO performance: the parameter $\beta$ and the quality of preference data. The existing literature has largely neglected the joint impact of these factors. This study investigates how varying $\beta$ and preference data quality influence DPO outcomes. It finds that optimal $\beta$ values depend on the informativeness of pairwise data. Based on this insight, the authors propose enhancements to DPO that involve batch-level dynamic $\beta$ calibration and $\beta$-guided data filtering. The efficacy of these improvements is empirically validated across two NLP tasks (dialogue and text summarization) using models of different sizes.

**Strengths:**

Overall, the paper is well-written and easy to follow.

The authors make an interesting empirical observation about DPO dynamics: an increase in $\beta$ improves performance when the gap in pairwise preference data is large but degrades performance when the gap is small.

The authors perform a thorough empirical analysis, including ablation and compatibility studies, to validate their proposed improvements.

The merits of the proposed enhancements to DPO are:
(1) Easy to implement, with no additional computational overhead.
(2) Compatible with other preference data filtering strategies and DPO variants like IPO and KTO.
(3) Utilizes the "running" reward discrepancy instead of relying on teacher/gold rewards.

**Weaknesses:**

Despite the comprehensive empirical analysis, the experiments are limited to specific datasets and model size ranges. Given the potential impact on the DPO literature, broader verification across diverse datasets and model sizes would strengthen the claims.

The dynamic $\beta$ approach could introduce instability by using the "running" reward discrepancy instead of teacher/gold rewards. A comparative analysis with methods that utilize teacher/gold rewards would be beneficial.

**Questions:**

In Figure 5a, including the $\beta$-guided filtering strategy on the x-axis alongside various gradient-based filtering strategies would be valuable. This addition would help assess its effectiveness compared to existing gradient-based approaches.

**Limitations:**

The authors acknowledge the limitations of their work in the conclusion. While the focus is on the technical improvement of preference alignment techniques for LLMs, a detailed discussion on the potential societal impact of these advancements would be a worthwhile addition.

---

> ### Author Rebuttal · Authors · 2024-08-07
>
> Thanks for your kind review. We are glad that you found our paper meaningful and easy to follow. We provide detailed answers to your comments below.
>
> **Q1: The experiments are limited to specific datasets and model size ranges.**
> > A1: Thank you for raising this concern. To expand our approach to more diverse datasets and model sizes, we follow the current state-of-the-art models SimPO[1]. We perform $\beta$-DPO with two families of models, Llama3-8B-Instruct and Mistral-7B-Instruct, on UltraChat-200k UltraFeedback. For comparison with baselines, we assess our models using one of the most popular open-ended instruction-following benchmarks: AlpacaEval 2. All settings are consistent with SimPO [1].
> > Please refer to Table below:
>
>
> | Method | Mistral-Instruct (7B) | Mistral-Instruct (7B) | Llama3-Instruct (8B) | Llama3-Instruct (8B) |
> |--------|----------------------|-----------------------|---------------------|---------------------|
> |        | LC (%)               | WR (%)                | LC (%)              | WR (%)              |
> | DPO    | 20.98                | **21.60**             | 40.44               | 37.38               |
> | $\beta$-DPO | **23.56**       | 20.42                 | **43.38**           | **38.21**           |
> | SimPO  | 28.50                | 30.56                 | 44.38               | 38.97               |
> | $\beta$-SimPO | **30.48**     | **32.13**             | **46.03**           | **40.18**           |
> > Table: AlpacaEval 2 results under the Mistral-Instruct (7B) and Llama3-Instruct (8B). LC and WR denote length-controlled and raw win rate, respectively. Regardless of whether we use Llama3-8B or Mistral-7B, and whether the loss function is DPO or SimPO, our $\beta$-{D, Sim}PO strategy consistently demonstrates significant performance improvements. This thoroughly showcases the method's strong generalization ability and excellent scalability.
>
> **Q2: Comparison with gold reward.**
> > A2: Thank you for pointing this out. In fact, computing the gold reward for the training set of 160k samples 1) greatly increases computational complexity and 2) makes the gold reward highly sensitive to the particular model used for annotation. Such an approach is difficult to generalize to arbitrary data scenarios.
> Additionally, to reduce the instability of "running" reward discrepancies, $\beta$-DPO utilizes batch-level calibration. Figure 5 (right) and Table 2 clearly demonstrate its superiority. This aligns with our original intent for this work: to develop a simple-to-implement, highly scalable dynamic $\beta$ strategy.
>
> **Q3: Including the $\beta$-guided filtering strategy.**
> > A3: Thank you for this suggestion. We have included the complete comparison chart in `REBUTTAL Figure 2 Left`. It can be clearly observed that: 1) the dynamic $\beta$ strategy can adapt to various data filtering methods, and 2) the $\beta$-guided filtering proposed in this paper remains optimal.
>
> [1] SimPO: Simple Preference Optimization with a Reference-Free Reward. Yu Meng and Mengzhou Xia and Danqi Chen.

---

> > ### Comment · Reviewer_3526 · 2024-08-11
> >
> > Thank you for the clarifications and additional experiments! I will increase my score.

---

### Official Review · Reviewer_fXTA · 2024-07-13

**Soundness:** 3
**Presentation:** 3
**Contribution:** 3
**Rating:** 6
**Confidence:** 4

**Summary:**

The authors's goal is to introduce adaptive schedules for the KL regularization $\beta$ in RLHF. This is a useful quality of life improvement with high potential impact on the final performance of an RLHF algorithm (similarly to how adaptive learning rate schedules are crucial in optimization).

As a guiding principle for the adaptive schedule the authors propose to adjust $\beta$ according to the "quality" of the data used in the update.  This decision is driven by preliminary experiments that show that:
- when RLHF is performed on low gap data (i.e. a human rater barely prefers a completion $y$ over $y'$ and there is no strong quality difference or preference towards $y$ or $y'$) the authors deem the data as high quality, and show that lowering the $\beta$ is beneficial as less regularization allows the model to fit the subtle difference between close pairs of examples
- when RLHF is performed on high gap data (i.e. a human rater strognly prefers a completion $y$ over $y'$) the authors deem the data as low quality, and show that increasing the $\beta$ is beneficial as more regularization prevents the model from overfitting only to the preferred sample at the detriment of pre-trained knowledge already stored in the weigths

Based on this quality principle, the author propose to measure the gap using a reward model to compute a reward discrepancy, and try to adjust $\beta$ as a function of the sample's reward discrepancy. This turns out to be too unstable and sensitive to outliers, so the authors propose a number of strategies to stabilize the learning while retaining an adaptive $\beta$
- adding a baseline reward discrepancy to "center" the $\beta$ updates
- moving from a per-sample reward discrepancy to batch statistics to smoothen the effect of outliers while retaining some adaptivity
- leveraging the reward discrepancy as a filtering measure, and simply skipping the updates on high-discrepancy (i.e. lowest quality) samples to reduce variance in the updates
- various combinations and ablations of the above

The authors show the usefulness of their approach on small to medium scale experiments using helpfulness and summarization text tasks, showing that an LLM judge prefers sentences from a policy fine-tuned with $\beta$-DPO to those from a DPO-finetuned policy, and in some cases even over the golden ground truth.

**Strengths:**

Adaptive schedules have always been pretty impactful in practice (although hard to tune), so I would consider the significance of this paper above average.

The principle used to quantify "quality" is both a strength and a weakness. The strength are:
- conceptually simple, and quite related to existing concept in the literature (e.g. class margins in regression as a similar quantity was used in SLiC [1r] ) which means there is literature and past insights that can guide both usage and future development
- computationally reasonable to compute (when using an implicit or explicit reward model) and in some cases even free (DPO implicit reward model, when the reward is part of the dataset)
- task-and-data-specific, so avoids a lot of common pitfalls that happen with e.g. schedules that depend only on features (e.g. maximum data norm) or not even that (e.g. linear warm-ups)

The paper is very clear and easy to follow.

Experiments include useful ablations, and in general are detailed and show improvement over baseline. A few aspects could be improved (more details in weaknesses)

**Weaknesses:**

The main (and critical weakness) of the paper is that while it extensively relies on a reward model (e.g. for computing reward discrepancies used in scheduling $\beta$ and filtering), I could not find details on how the authors recommend this reward model is implemented in practice. The two ways I can imagine.

The first is to pre-train a reward model as described in section 3 (e.g. by fitting the model in Eq. 2). If this is the case there are a bunch of disconnects in the paper:
- this disconnects the approach from DPO (which builds its own implicit reward model), resulting in different reward estimates for the tuning of $\beta$ and for the PO procedure
- if the reward model is trained on the same offline data used in the $\beta$-DPO run, then negative reward discrepancies should be treated differently than positive discrepancies, since r(y_w) < r(y_l) indicates either an outlier but possibly also inaccuracy of the reward model (and in general should not happen as often as described by the experiments)

The second is to use directly the implicit reward model induced by the policy trained by DPO:
- this solves the disconnect between DPO and discrepancy rewards, but note that the accuracy of the intermediate implicit models is bound to be poor, as the goal of DPO is to start with a poor policy (poor reward model) and improve it over time, relying on the supervised signal of offline winners and losers (which does not depend on neither policy or reward model) to drive this improvements. as a consequence using the intermediate rewards for additional online choices (tuning $\beta$, filtering) might be not very sound in theory (but maybe works in practice?)
- an implicit DPO reward model makes the whole process even more non-stationary

At an empirical level, while the combination of dynamic schedule and filtering proves effective, it is unclear when only one or both techniques are necessary. This weakens the contribution as $\beta$-DPO is not only DPO+a dynamic schedule but DPO+dynamic schedule + filtering. As such, in the natural attempt to transfer the schedule to other PO methods (e.g. KTO, IPO, c-DPO, RPO, etc..) one must also find a way to transfer the filtering technique, making the contribution less general.

The principle used to quantify "quality" is both a strength and a weakness. The weaknesses are:
- the guidelines proposed that connect $\beta$ schedule to reward gap (high gap -> high $\beta$, low gap -> low $\beta$) seem entirely driven by empirical observations in Figure 1, and are not supported by any further insight or justifications for why this might be the one correct choice when moving away from a constant schedule. More insight would be valuable, where for example DPO gradients are rescaled by a $(1 + e^{M_i})^{-1}$ which means that $\beta$ is somewhat rescaled according to the gradient norm, although in a non-monotonic way. While the proposed dynamic schedule might sound reasonable (or at least I found it so), the exact opposite (high gap -> low $\beta$ since the high margin gives us confidence to fit fully, low gap -> high $\beta$ to avoid overfitting random perturbations on the reward) also sounds reasonable. The soundness of this "quality" measure and its role in tuning $\beta$ should be either evaluated on ablations against other dynamic schedules/principles as baslines, or supported theoretically before it can be fully considered a strong contribution.
- according to the authors introduction, the "quality" has a (at least) bimodal distribution in the dataset (high gap data and low gap data), however outlier filtering is performed using only an unimodal 3sigma principle, which looks counterintuitive to me (e.g., aren't all high gaps pairs technically outliers from the low gap sample point of view and vice-versa?)

Experiments are good, but without better explainations on how the reward is computed they are hard to evaluate.

**Questions:**

How is the reward model constructed and used? (this is my main question and I am open to revise my score based on the answer)

Can you strengthen your support for the gap principle beyond the empirical observation of Figure 1?

It seems pretty convincing that a dynamic schedule outperforms a fixed one. There is a missing ablation showing that the one you proposed is strongly preferrable over others (e.g., high gap -> low $\beta$, low gap -> high $\beta$). Can you show that different schedule do not work in your experiments (adapting the filtering if necessary)? Alternatively, can you show which schedules work for which datasets (while keeping that in general dynamic schedules are better than a fixed schedule)?

**Limitations:**

The main limitations as I see now are the lack of clarity on the reward model, and the limited support on why this specific schedule is to be preferred over other dynamic schedules.

Beyond those, the authors correctly identify important other open questions in the limitation settings. I'd like to highlight among those two that I think would be very relevant in increasing the impact of the paper and make it jump several levels in my evaluation:
- Testing the approach beyond DPO. Just like learning rate schedules generalize across optimizers, it's crucial for a good $\beta$ dynamic schedule to generalize across PO methods like c-DPO/IPO/SPO/RPO etc...
- Automated parameter tuning. The authors make a good effort on this aspect (automatic choice for $\beta$ and variance estimation) with only one hyperparameter left ($M_0$) to tune despite moving from a simple constant schedule to a dynamic one. However, a true hyperparameter-free method would be much more valuable.

---

> ### Author Rebuttal · Authors · 2024-08-07
>
> Dear Reviewer,
>
> Thanks for your kind review. We are glad that you found our paper clear and easy to follow. We provide detailed answers to your comments below.
>
> **Q1: How is the reward model constructed and used?**
> > A1: Thank you for pointing this out. **We directly use the implicit reward model induced by the policy trained by DPO, where the reward discrepancy in DPO is expressed as: $ \beta \log (\frac{\pi\_\theta (y\_w \mid x) }{\pi\_{\text{ref}}(y\_w \mid x)}) - \beta \log (\frac{\pi\_\theta (y\_l \mid x) }{\pi\_{\text{ref}}(y\_l \mid x)})$.**
>
> **Q1.1: As a consequence, using intermediate rewards for additional online choices (such as tuning and filtering) may lack theoretical soundness.**
> > **A1.1:** In the early stages of training, $\beta$-DPO behaves similarly to DPO (see `REBUTTAL Figure 2 Middle`). Due to the low accuracy of the intermediate implicit models, the margin between winners and losers is minor, resulting in $\beta\_{\text{batch}} \approx \beta\_0$. However, in the later stages of training, as the margin begins to exhibit significant differences, the advantages of a dynamic $\beta$ become apparent.
>
> **Q1.2: An implicit DPO reward model introduces greater non-stationarity into the process.**
> > **A1.2:** This question raises a pertinent point. An implicit DPO reward model can lead to instability in the estimation of dynamic $\beta$, thus making the entire training process more non-stationary. To address this issue, we propose that batch-level calibration is essential. As demonstrated in Table 2 of the initial manuscript, instance-level calibration on dynamic $\beta$ results in a substantial performance drop. In contrast, batch-level calibration enhances stationarity and accentuates the effects of dynamic $\beta$. Additionally, Figure 5 (Right) illustrates that instance-level calibration exacerbates the influence of outliers.
>
> **Q2: It is unclear when only one or both techniques are necessary.**
> > A2: As demonstrated in Table 1 of the original manuscript, we observe that in a smaller model (410M), the improvement of data filtering is more significant, while in a larger model (2.8B), the improvement of dynamic $\beta$ is more significant. Additionally, the dynamic schedule can also improve other filtering methods (Figure 5 Left). **Since our filtering method relies only on implicit reward discrepancy, it also works with other PO methods (Figure 5 Middle).**
>
> **Q3: Can you strengthen your support for the gap principle beyond the empirical observation of Figure 1?**
> > A3: We also attempt the exact opposite (high gap -> low $\beta$; low gap -> high $\beta$). Refer to `REBUTTAL Figure 1, REBUTTAL Figure 2 Right` for the experimental results. We find that across multiple datasets and models, the current strategy remains optimal.
>
> **Q4: Outlier filtering is performed using only an unimodal 3-sigma principle**
> > A4: Utilizing the 3-sigma principle is primarily to reduce the bias in $\beta$ estimation and thus enhance training stability. Intuitively, high-gap samples often carry low information content, whereas low-gap samples might still be noisy, indicating that winners and losers may not be well distinguished (label flipping). We hope this helps you gain an intuitive understanding. A more detailed analysis can be found in the original manuscript, Section 4.1 (lines 157-170).
>
> **Q5: Testing the approach beyond DPO**
> >A5: Thank you for pointing this out. **First, we demonstrate Dynamic $\beta$ Enhancement across IPO, KTO, and SPPO in Figure 5 (Middle) of the original paper.**
> To expand our approach to more diverse datasets and model sizes, we follow the current state-of-the-art models SimPO[1], which surpass c-DPO/RPO etc. We perform $\beta$-DPO with two families of models, Llama3-8B-Instruct and Mistral-7B-Instruct, on UltraChat-200k UltraFeedback. For comparison with baselines, we assess our models using one of the most popular open-ended instruction-following benchmarks: AlpacaEval 2. All settings are consistent with SimPO [1].
> > Please refer to `REBUTTAL Table 1`. Regardless of whether we use Llama3-8B or Mistral-7B, and whether the loss function is DPO or SimPO, our $\beta$-{D, Sim}PO strategy consistently demonstrates significant performance improvements. This thoroughly showcases the method's strong generalization ability and excellent scalability.
>
>
> **Q6: Automated parameter tuning.**
> > A6: **In most scenarios, setting $\alpha = \frac{2}{M_0}$ yields stable performance improvements, where $M_0$ can be estimated using a moving average updating scheme (refer to Equation 7).** This is informed by the formula $\beta\_{\text{batch}} = [1 + \alpha(\mathbb{E}\_{i \sim \text{batch}}[M\_i] - M\_0)]\beta\_0$, resulting in an overall change range of $[\frac{2\mathbb{E}\_{i \sim \text{batch}}[M\_i] - M\_0}{M\_0}]\beta\_0$, which normalizes based on $M\_0$ over the foundation of $\beta\_0$.
>
> | | HH | TLDR |
> |-------------------|-------------------|-------------|
> | DPO               | 51.01               |   32.45       |
> | $\beta$-DPO               | 57.68              | 51.67|
> | $\beta$-DPO ($\frac{2}{M\_0}$)              | 58.02               | 51.32 |
>
> > To substantiate this perspective, we present performance in the above table, demonstrating that our setting achieves significant enhancements across various datasets and models compared to DPO, **without imposing additional pressure for hyperparameter searches.** We appreciate your concern; while we believe that further theoretical consolidation is a meaningful future endeavor, we maintain that the $\beta$-DPO approach remains valuable, offering a straightforward (not overly reliant on hyperparameter tuning) and effective (stable performance enhancements) new paradigm for fine-tuning large models and studying data quality.
>
> [1] SimPO: Simple Preference Optimization with a Reference-Free Reward. Yu Meng and Mengzhou Xia and Danqi Chen.

---

> ### Author Response · Authors · 2024-08-13
> **Thank you for your time and consideration. We look forward to hearing back from you.**
>
> Dear reviewer,
>
> We greatly appreciate your invaluable feedback. We now aim to provide a concise summary that carefully addresses your main concerns. We hope that this effort will be worthy of your support.
>
> **Q1: How is the reward model constructed and used?**
> > **A1:** As you mentioned in your second point, **we directly utilize the implicit reward model induced by the policy trained with DPO, where the reward discrepancy in DPO is expressed as: $ \beta \log (\frac{\pi_\theta (y_w \mid x) }{\pi_{\text{ref}}(y_w \mid x)}) - \beta \log (\frac{\pi_\theta (y_l \mid x) }{\pi_{\text{ref}}(y_l \mid x)})$.**
>
> **Q1.1: As a consequence, using intermediate rewards for additional online choices (such as tuning and filtering) may lack theoretical soundness.**
> > **A1.1:** In the early stages of training, $\beta$-DPO exhibits similar behavior to DPO (`REBUTTAL Figure 2 Middle`). Due to the low accuracy of the intermediate implicit models, the margin between winners and losers is small, resulting in $\beta_{\text{batch}} \approx \beta_0$. However, as training progresses and the margin starts to show significant differences, the benefits of a dynamic beta become evident.
>
> **Q1.2: An implicit DPO reward model introduces greater non-stationarity into the process.**
> > **A1.2:** An implicit DPO reward model can indeed lead to instability in the estimation of dynamic $\beta$, making the entire training process more non-stationary. To mitigate this issue, we propose that batch-level calibration is crucial. As shown in Table 2 of the original manuscript, instance-level calibration on dynamic $\beta$ leads to a significant performance drop. In contrast, batch-level calibration improves stationarity and emphasizes the effects of dynamic $\beta$.
>
> **Furthermore, the moving average updating scheme on $M_0$ (ref to Equation 7) helps alleviate the impact of （`Q1.1`） the poor reward model and （`Q1.2`） potential instability.** Although the reward margin increases further ($M_i$ increases) as the model's capability improves, $\beta_{\text{Batch}}$ is positively correlated with $M_i - M_0$. Consequently, $\beta$-DPO focuses more on the dynamic reward of different datapoints relative to global datapoints, **rather than their absolute values.** This approach may help reduce the impact of the changing reward margins on the overall stability of the system. The evolution of $\beta_{\text{Batch}}$ over the course of training steps, as illustrated in `REBUTTAL Figure 2 Middle`, supports this perspective.
>
> We were wondering if our responses have addressed your concerns since the discussion phase is coming to a close. We are also eager to know if you have any other concerns or suggestions. Thank you for your time and consideration!

---

> > ### Author Response · Authors · 2024-08-14
> > **Inquiry on Additional Feedback**
> >
> > Thanks for your constructive feedback on our paper. We kindly inquire whether there may exist any additional concerns or unresolved questions that might be impeding the paper's attainment of a higher rating. We are available for any further clarifications or discussions!

---

### Author Rebuttal · Authors · 2024-08-07

We thank all reviewers for their valuable and insightful feedback.
- We are encouraged that the reviewers found our paper meaningful (Reviewers $\color{red}{\text{fXTA}}$, $\color{green}{\text{UzDb}}$, $\color{black}{\text{4RgK}}$).
- Moreover, we are grateful that the reviewers found our proposed $\beta$-DPO algorithm simple and effective (Reviewers $\color{blue}{\text{3526}}$, $\color{green}{\text{UzDb}}$, $\color{orange}{\text{uGzu}}$).
- We also appreciate that the reviewers found our paper easy to follow and well-written (Reviewers $\color{red}{\text{fXTA}}$, $\color{blue}{\text{3526}}$, $\color{black}{\text{4RgK}}$).



We also appreciate reviewers pointing out our weaknesses. We address their comments point by point and try our best to respond to them. We hope our response addresses the reviewers' concerns.

The additional experiments in the Rebuttal PDF are summarised as follows:

- In `Rebuttal Figure 1`, we compare the different dynamic schedules on HH and TLDR.
- In `Rebuttal Figure 2 (Left)`, we introduce the $\beta$-guided filtering strategy.
- In `Rebuttal Figure 2 (Middle)`, we visualize the range of beta in experiments and the corresponding preference accuracy across training steps.
- In `Rebuttal Figure 2 (Right)`, we compare the different dynamic schedules on AlpacaEval.
- In `Rebuttal Table 1`, we extend our approach to include a more diverse set of datasets and model sizes.
- In `Rebuttal Table 2`, we conduct a parameter sensitivity analysis on $m$.

We have carefully considered the comments and suggestions provided by the reviewers, and we have addressed them point by point in our rebuttal. We believe that our responses adequately address the concerns raised.

Once again, we sincerely thank the reviewers for their valuable feedback, which has significantly contributed to the improvement of our work.

---

> ### Author Response · Authors · 2024-08-14
> **New Global Response by Authors**
>
> First of all, we would like to express our sincere gratitude to all reviewers for their time and efforts in evaluating our submission. As the Author-Reviewer discussion period is drawing to a close, we would like to summarize the discussion results and the improvements made to our paper.
>
> Below, we summarize our discussions and feedback with the reviewers:
>
> - Reviewer $\color{red}{\text{fXTA}}$: We have provided a detailed explanation of the implicit RM expression used in this paper and how it mitigates the corresponding non-stationarity. We believe that the extensive comparative experiments are sufficient to validate the reliability of our method. Thank you for your constructive and valuable comments.
>
> - Reviewer $\color{blue}{\text{3526}}$: We have conducted experiments on more diverse datasets and model sizes to further validate the effectiveness of $\beta$-DPO. Considering your positive rating, we believe we have addressed your concerns. Thank you for your positive feedback.
>
> - Reviewer $\color{green}{\text{UzDb}}$: We have clarified the implementation of the reward model and further explained why low-gap examples are considered high-quality. Additionally, we have attempted to use an explicit RM to strengthen the claim that the method is "reward-model independent". We believe our rebuttal has addressed your concerns, considering that you have further improved your ratings of our work.
>
> - Reviewer $\color{orange}{\text{uGzu}}$: We have conducted experiments on more diverse datasets and model sizes, and visualized the range of values for $\beta_{\text{Batch}}$. Considering your positive rating, we believe we have addressed your concerns. Thank you for your positive feedback.
>
> - Reviewer $\color{black}{\text{4RgK}}$: We have further analyzed the impact of different model sizes, automated parameter tuning, batch sizes, and learning rates. Thank you for your constructive and valuable comments.
>
> We are pleased to note that Reviewers $\color{blue}{\text{3526}}$, $\color{green}{\text{UzDb}}$, and $\color{orange}{\text{uGzu}}$ have provided positive ratings and their concerns have been addressed. For Reviewers $\color{red}{\text{fXTA}}$ and $\color{black}{\text{4RgK}}$, we understand that you may have been too busy to participate in the discussion phase. However, we kindly request that you re-evaluate our paper in the subsequent stages.
>
> As the deadline for the author-reviewer discussion approaches, we sincerely hope that our efforts and improvements will be taken into consideration. Once again, we thank the reviewers for their efforts in helping to improve the quality of our paper.

---

### Public Comment · ~Heyang_Gong1 · 2025-01-02
**adaptive beta parameter?**

是否有考虑将 $\beta(x)$ 视为输入 $x$ 的函数，建模为一个可学习的动态参数? 我的疑问是你手动去设计这个基于奖励参数差异的, 这确实或许带来了稳定性，但是也带来了偏差呀? 我们为什么不考虑一下  instance 级别 beta function + 一些降低方差的正则呢?

---

> ### Public Comment · ~Junkang_Wu1 · 2025-04-15
> **感谢关注**
>
> 您好，
>
> 感谢您的来信以及对我们的工作提出的宝贵建议。关于参数化 $\beta(x)$ 的方法，虽然确实是一种可行的解决方案，但在实际调节过程中，此类函数以及正则项仍存在诸多不确定性和不稳定性。例如，$\beta(x)$ 取值过大或过小可能导致效果不佳，因此仍需依赖较多手动调整。本研究的主要目标是提出动态化的设计思想，并展示其良好的泛化性能。更多的改进空间和优化方向，我们期待未来研究能够进一步探索！
>
> 祝好，
> Junkang
>
> ---
>
> Dear Heyang,
>
> Thank you for your email and the valuable suggestions regarding our work. While parameterizing $\beta(x)$ is indeed a feasible approach, tuning such functions along with regularization terms still involves significant uncertainties and instabilities in practice. For instance, an excessively large or small $\beta(x)$ may lead to suboptimal results, necessitating considerable manual intervention. The primary goal of this study is to propose the dynamic design concept and demonstrate its robust generalization performance. We believe further improvements and refinements remain open questions for future research!
>
> Best regards,
> Junkang

---

### Decision · Program_Chairs · 2024-09-25

**Decision:**

Accept (poster)

**Comment:**

This paper is an interesting first contribution toward the idea of adaptively tuning the amount of regularization in DPO. The authors show that their method of adaptively changing the regularization parameter $\beta$ (for KL-divergence based regularization) combined with an approach for filtering examples is able to significantly boost the performance of many implicit reward model-based approaches including DPO. In terms of theoretical understanding, this work seems quite preliminary. Answering in a principled way the question of whether $\beta$ should be smaller or larger in the case of a small gap (respectively large gap), and getting a more theoretically principled understanding for how to change $\beta$, would be valuable. However, the empirical results are promising, and the intuition provided thus far is a fair (not strong, but fair) start.

The paper overall reads clearly. Yet, multiple well-informed reviewers came away with a very large misunderstanding of the paper: originally, some reviewers thought the authors were using an external reward model, and the authors needed to clarify that in fact they are using an implicit reward model (the one from DPO). After the authors provided this clarification (and in one case, showed additional experiments with external reward models), the reviewers were positive about this paper. I believe the misunderstanding arose due to the writing of the paper, and so I stress that the authors should be as clear as possible about the reward model when revising their work.

One reviewer who was rather negative on this work took issue with the paper having multiple, new hyperparameters. Some of the authors’ responses here seem fair, but there is a risk that researcher degrees of freedom may have played a role in the results of the new techniques being better. Yet, given the consistent improvement seen for multiple implicit reward model-based approaches, several datasets in the original paper, and the new experiments provided during the discussion phase, I believe the above-mentioned risk is low.

Another issue was that the original submission didn’t have sufficiently diverse experiments (in terms of datasets and size of the model). This point was well-addressed in the discussion period with the authors.

Overall, this is a fine piece of work. I believe it is at the threshold of publication at NeurIPS, and it may spark very interesting follow-up works that lead to better theoretical understanding and also even better ways of doing dynamic regularization.